# Skill-aware Mutual Information Optimisation for Generalisation in Reinforcement Learning

**Xuehui Yu**[1,2]    **Mhairi Dunion**[2]    **Xin Li**[1]    **Stefano V. Albrecht**[2]
[1]Harbin Institute of Technology    [2]University of Edinburgh
{yuxuehui,22s103169}@stu.hit.edu.cn
{mhairi.dunion,s.albrecht}@ed.ac.uk

## Abstract

Meta-Reinforcement Learning (Meta-RL) agents can struggle to operate across tasks with varying environmental features that require different optimal *skills* (i.e., different modes of behaviour). Using context encoders based on contrastive learning to enhance the generalisability of Meta-RL agents is now widely studied but faces challenges such as the requirement for a large sample size, also referred to as the $\log$-$K$ curse. To improve RL generalisation to different tasks, we first introduce **S**kill-**a**ware **M**utual **I**nformation (SaMI), an optimisation objective that aids in distinguishing context embeddings according to skills, thereby equipping RL agents with the ability to identify and execute different skills across tasks. We then propose **S**kill-**a**ware **N**oise **C**ontrastive **E**stimation (SaNCE), a $K$-sample estimator used to optimise the SaMI objective. We provide a framework for equipping an RL agent with SaNCE in practice and conduct experimental validation on modified MuJoCo and Panda-gym benchmarks. We empirically find that RL agents that learn by maximising SaMI achieve substantially improved zero-shot generalisation to unseen tasks. Additionally, the context encoder trained with SaNCE demonstrates greater robustness to a reduction in the number of available samples, thus possessing the potential to overcome the $\log$-$K$ curse.

## 1 Introduction

Reinforcement Learning (RL) agents often learn policies that do not generalise across tasks in which the environmental features and optimal *skills* are different [des Combes et al., 2018, Garcin et al., 2024]. Consider a set of cube-moving tasks where an agent is required to move a cube to a goal position on a table (Figure 1). These tasks become challenging if environmental features, such as table friction, vary between tasks. When facing an unknown environment, the agent

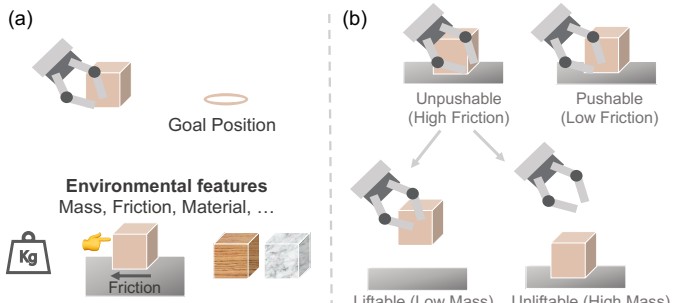

Figure 1: (a) In a cube-moving environment, tasks are defined according to different environmental features. (b) Different tasks have different transition dynamics caused by underlying environmental features, hence optimal skills are different across tasks.

needs to explore effectively, understand the environment, and adjust its behaviour accordingly within an episode. For instance, if the agent tries to push a cube across a table covered by a tablecloth and finds it "unpushable," it should infer that the table friction is relatively high and adapt by lifting

38th Conference on Neural Information Processing Systems (NeurIPS 2024).

the cube to avoid friction, rather than continuing to push. Recent advances in Meta-Reinforcement Learning (Meta-RL) [Lee et al., 2020, Agarwal et al., 2021, Mu et al., 2022, Dunion et al., 2023b,a, McInroe et al., 2024] enable agents to understand environmental features by inferring context embeddings from a small amount of exploration, and to train a policy conditioned on the context embedding to generalise to novel tasks.

In recent years, unsupervised contrastive learning algorithms have been shown to learn context embeddings that perform remarkably well on generalisation tasks [Clavera et al., 2019a, Lee et al., 2020]. In particular, some Meta-RL algorithms [Fu et al., 2021, Wang et al., 2021, Li et al., 2021, Sang et al., 2022] train context encoders by maximising InfoNCE [Oord et al., 2019], which has yielded vital insights into contrastive learning through the lens of mutual information (MI) analysis. The InfoNCE objective can be interpreted as a $K$-sample lower bound on the MI between trajectories and context embeddings. Despite significant advances, integrating contrastive learning with Meta-RL poses several unresolved challenges, of which two are particularly relevant to this research: **(i) Existing context encoders based on contrastive learning do not distinguish tasks that require different skills**. Many prior algorithms only pull embeddings of the same tasks together and push those of different tasks apart. However, for example, a series of cube-moving tasks with high friction may only require a Pick&Place skill (picking the cube off the table and placing it at the goal position), making further differentiation unnecessary. **(ii) Existing $K$-sample MI estimators are sensitive to the sample size $K$ (i.e., the log-$K$ curse)** [Poole et al., 2019]. The low sample efficiency of RL [Franke et al., 2021] and the sample limitations in zero-shot generalisation make collecting a substantial quantity of samples often impractical [Arora et al., 2019, Nozawa and Sato, 2021]. The effectiveness of $K$-sample MI estimators breaks down with a finite sample size and leads to a significant performance drop in downstream RL tasks [Mnih and Teh, 2012, Guo et al., 2022].

To enhance RL generalisation across different tasks, we propose that the context embeddings should optimise downstream tasks and indicate whether the current skill remains optimal or requires further exploration. This also reduces the necessary sample size by focusing solely on skill-related information. In this work, (1) we introduce *Skill-aware Mutual Information (SaMI)*, a generalised form of MI objective between context embeddings, skills, and trajectories, designed to address issue (i). We provide a theoretical proof showing that by introducing skills as a third variable into the MI of context embeddings and trajectories, the resulting SaMI is smaller and easier to optimise. Furthermore, (2) we propose a data-efficient $K$-sample estimator, *Skill-aware Noise Contrastive Estimation (SaNCE)* to optimise SaMI, effectively addressing issue (ii). Additionally, (3) we propose a practical skill-aware trajectory sampling strategy that shows how to sample positive and negative examples without relying on any prior skill distribution. In that way, Meta-RL agents autonomously acquire a set of skills applicable to many tasks, with these skills emerging solely from the SaMI learning objective and data.

We demonstrate empirically in MuJoCo [Todorov et al., 2012] and Panda-gym [Gallouédec et al., 2021] that SaMI enhances the zero-shot generalisation capabilities of two Meta-RL algorithms [Yu et al., 2020, Fu et al., 2021] by achieving higher returns and success rates on previously unseen tasks, ranging from moderate to extreme difficulty. Visualisation of the learned context embeddings reveals distinct clusters corresponding to different skills, suggesting that the SaMI learning objective enables the context encoder to capture skill-related information from trajectories and incentivise Meta-RL agents to acquire a diverse set of skills. Moreover, SaNCE enables Meta-RL algorithms to use smaller sample spaces while achieving improved downstream control performance, indicating their potential to overcome the log-$K$ curse.

## 2 Related works

**Meta-RL.** By conditioning on an effective context embedding, Meta-RL policies can zero-shot generalise to new tasks with a small amount of exploration Kirk et al. [2023]. Existing algorithms can be categorised into three types based on different context embeddings. In the first category, the context embedding is learned by minimising the downstream RL loss [Rakelly et al., 2019, Yu et al., 2020]. PEARL [Rakelly et al., 2019] learns probabilistic context embeddings by recovering the value function. Multi-task SAC with task embeddings as inputs to policies (TESAC) [Hausman et al., 2018, Yu et al., 2020] parameterises the learned policies through a shared embedding space, aiming to maximise the average returns across tasks. However, the update signals from the RL loss are stochastic and weak, and may not capture the similarity relations among tasks [Fu et al.,

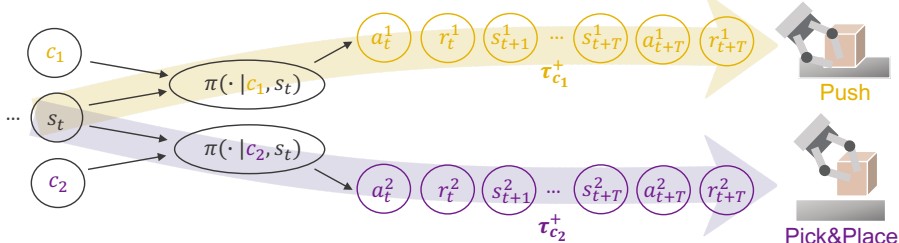

Figure 2: A policy $\pi$ conditioned on a fixed context embedding $c$ is defined as a skill $\pi(\cdot|c)$ (shortened as $\pi_c$). The policy $\pi$ conditioned on a fixed $c$ alters the state of the environment in a consistent way, thereby exhibiting a mode of skill. The skill $\pi(\cdot|c_1)$ moves the cube on the table in trajectory $\tau_{c_1}^+$ and is referred to as the Push skill; correspondingly, the Pick&Place skill $\pi(\cdot|c_2)$ takes the cube off the table and places it in the goal position in the trajectory $\tau_{c_2}^+$.

2021]. The second category involves learning context embeddings through dynamics prediction [Lee et al., 2020, Zhou et al., 2019], which can make the context embeddings noisy, as they may model irrelevant dependencies and overlook task-specific information [Fu et al., 2021]. The third category employs contrastive learning [Fu et al., 2021, Wang et al., 2021, Li et al., 2021, Sang et al., 2022], achieving significant improvements in context learning. However, these methods overlook the similarity of skills between different tasks, thus failing to achieve effective zero-shot generalisation by executing different skills. Our improvements build upon this third category by distinguishing context embeddings according to different optimal skills.

**Contrastive learning.** Contrastive learning has been applied to RL due to its significant momentum in representation learning in recent years, attributed to its superior effectiveness [Tishby and Zaslavsky, 2015, Hjelm et al., 2019, Dunion and Albrecht, 2024], ease of implementation [Oord et al., 2019], and strong theoretical connection to MI estimation [Poole et al., 2019]. MI is often estimated using InfoNCE [Oord et al., 2019] that has gained recent attention due to its lower variance [Song and Ermon, 2020] and superior performance in downstream tasks. However, InfoNCE may underestimate the true MI when the sample size $K$ is finite. To address this limitation, CCM [Fu et al., 2021] uses a large number of samples for maximising InfoNCE. DOMINO [Mu et al., 2022] reduces the true MI by introducing an independence assumption; however, this results in biased estimates. We focus on proposing an unbiased alternative MI objective and a more data-efficient $K$-sample estimator tailored for downstream RL tasks, which, to our knowledge, have not been addressed in previous research.

## 3 Preliminaries

**Reinforcement learning.** In Meta-RL, we assume an *environment* is a distribution $\xi(e)$ of *tasks* $e$ (e.g. uniform in our experiments). Each task $e \sim \xi(e)$ has a similar structure that corresponds to a Markov Decision Process (MDP) [Puterman, 2014], defined by $\mathcal{M}_e = (\mathcal{S}, \mathcal{A}, R, P_e, \gamma)$, with a state space $\mathcal{S}$, an action space $\mathcal{A}$, a reward function $R(s_t, a_t)$ where $s_t \in \mathcal{S}$ and $a_t \in \mathcal{A}$, state transition dynamics $P_e(s_{t+1}|s_t, a_t)$, and a discount factor $\gamma \in [0, 1)$. In order to address the problem of zero-shot generalisation, we consider the transition dynamics $P_e(s_{t+1}|s_t, a_t)$ vary across tasks $e \sim \xi(e)$ according to multiple *environmental features* $e = \{e^0, e^1, ..., e^N\}$ that are not included in states $s$ and can be continuous random variables, such as mass and friction, or discrete random variables, such as the cube's material. For instance, in a cube-moving environment (Figure 1), an agent has different tasks that are defined by different environmental features (e.g., mass and friction). The Meta-RL agent's goal is to learn a generalisable policy $\pi$ that is robust to such dynamic changes. Specifically, given a set of training tasks $e$ sampled from $\xi_{\text{train}}(e)$, we aim to learn a policy that can maximise the discounted returns, $\arg\max_\pi \mathbb{E}_{e \sim \xi_{\text{train}}(e)}[\sum_{t=0}^{\infty} \gamma^t R(s_t, a_t)|a_t \sim \pi(a_t|s_t), s_{t+1} \sim P_e(s_{t+1}|s_t, a_t)]$, and can produce accurate control for unseen test tasks sampled from $\xi_{\text{test}}(e)$.

**Contrastive learning.** In Meta-RL, the context encoder $\psi(c|\tau_{c,0:t})$ first takes the trajectory $\tau_{c,0:t} = \{s_0, a_0, r_0, ..., s_t\}$ from the current episode as input and compresses it into a context embedding $c$ [Fu et al., 2021]. Then, the policy $\pi$, conditioned on context embedding $c$, consumes the current state $s_t$ and outputs the action $a_t$. As a key component, the quality of context embedding $c$ can affect algorithms' performance significantly. MI is an effective measure of embedding quality [Goldfeld

et al., 2019], hence we focus on a context encoder that optimises the InfoNCE objective $I_{\text{InfoNCE}}(x; y)$, which is a $K$-sample estimator and lower bound of the MI $I(x; y)$ [Oord et al., 2019]. Given a query $x$ and a set $Y = \{y_1, ..., y_K\}$ of $K$ random samples containing one positive sample $y_1$ and $K - 1$ negative samples from the distribution $p(y)$, $I_{\text{InfoNCE}}(x; y)$ is obtained by comparing pairs sampled from the joint distribution $x, y_1 \sim p(x, y)$ to pairs $x, y_k$ built using a set of negative examples $y_{2:K}$:

$$I_{\text{InfoNCE}}(x; y|\psi, K) = \mathbb{E}\left[\log \frac{f_\psi(x, y_1)}{\frac{1}{K}\sum_{k=1}^{K} f_\psi(x, y_k)}\right]. \tag{1}$$

InfoNCE constructs a formal lower bound on the MI, i.e., $I_{\text{InfoNCE}}(x; y|\psi, K) \leq I(x; y)$ [Guo et al., 2022, Chen et al., 2021]. Given two inputs $x$ and $y$, their *embedding similarity* is $f_\psi(x, y) = e^{\psi(x)^\top \cdot \psi(y)/\beta}$, where $\psi$ is the context encoder that projects $x$ and $y$ into the context embedding space, the dot product is used to calculate the similarity score between $\psi(x), \psi(y)$ pairs [Wu et al., 2018, He et al., 2020], and $\beta$ is a temperature hyperparameter that controls the sensitivity of the product. Some previous Meta-RL methods [Lee et al., 2020, Mu et al., 2022] learn a context embedding $c$ by maximising $I_{\text{InfoNCE}}(c; \tau_c|\psi, K)$ between the context $c$ embedded from a trajectory in the current task, and the historical trajectories $\tau_c$ under the same environmental features setting.

## 4 Skill-aware mutual information optimisation for Meta-RL

### 4.1 The log-$K$ curse of $K$-sample MI estimators

In this section, we provide a theoretical analysis of the challenge inherent in learning a $K$-sample estimator for MI, commonly referred to as the log-$K$ curse. Based on this theoretical analysis, we give insights to overcome this challenge. Given that we focus on the generalisation of RL, we only consider cases with a finite sample size of $K$. If a context encoder $\psi$ in Equation (1) has sufficient training epochs, then $I_{\text{InfoNCE}}(x; y|\psi, K) \approx \log K$ [Mnih and Teh, 2012, Guo et al., 2022]. Hence, the MI we can optimise is bottlenecked by the number of available samples, formally expressed as:

**Lemma 1** *Learning a context encoder $\psi$ with a $K$-sample estimator and finite sample size $K$, we have* $I_{\text{InfoNCE}}(x; y|\psi, K) \leq \log K \leq I(x; y)$, *when $x \not\!\perp\!\!\!\perp y$ (see proof in Appendix A).*

We do not consider the case when $x \perp\!\!\!\perp y$, i.e., $\log K \geq I(x; y) = 0$ ($\forall K \geq 1$), because a Meta-RL agent learns a context encoder by maximising MI between trajectories $\tau_c$ and context embeddings $c$, which are not independent (as shown in Figure 2). While an unbounded sample size $K$ for learning effective context embeddings is theoretically feasible and assumed in many studies Mu et al. [2022], Lee et al. [2020], it is often impractical in practice. Therefore, building on Lemma 1, we derive two key insights with limited samples: (1) generalising the current MI objective to be smaller than $I(x; y)$ (see Section 4.2); (2) developing a $K$-sample estimator tighter than $I_{\text{InfoNCE}}$ (see Section 4.3).

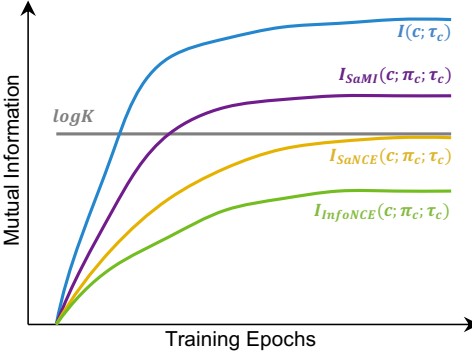

Figure 3: $I_{\text{InfoNCE}(c; \pi_c; \tau_c)}$, with a finite sample size of $K$, is a loose lower bound of $I(c; \tau_c)$ and leads to lower performance embeddings. $I_{\text{SaMI}}(c; \pi_c; \tau_c)$ is a lower ground-truth MI, and $I_{\text{SaNCE}}(c; \pi_c; \tau_c)$ is a tighter lower bound.

### 4.2 Skill-aware mutual information: a smaller ground-truth MI

We aim for our MI learning objective to incentivise agents to acquire a diverse set of skills, enabling them to generalise effectively across tasks. To start with, we define skills [Eysenbach et al., 2018]:

**Definition 1 (Skills)** *A policy $\pi$ conditioned on a fixed context embedding $c$ is defined as a skill $\pi(\cdot|c)$, abbreviated as $\pi_c$. If a skill $\pi_c$ is conditioned on a state $s_t$, we can sample actions $a_t \sim \pi(\cdot|c, s_t)$. After sampling actions from $\pi_c$ at consecutive timesteps, we obtain a trajectory $\tau_{c,t:t_T} = \{s_t, a_t, r_t, s_{t+1}, \ldots, s_{t+T}, a_{t+T}, r_{t+T}\}$ which demonstrates a consistent mode of behaviour.*

After a limited amount of exploration, an agent should be able to infer the task (i.e., environmental features $e = e^0, e^1, \ldots, e^N$) and adapt accordingly within the current episode. The context em-

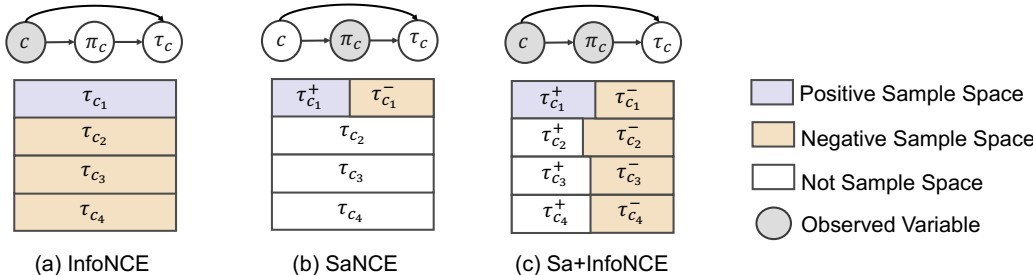

Figure 4: A comparison of sample spaces for task $e_1$. Positive samples $\tau_{c_1}$ or $\tau_{c_1}^+$ are always from current task $e_1$. For SaNCE, in a task $e_k$ with embedding $c_k$, the positive skill $\pi_{c_k}^+$ conditions on $c_k$ and generates positive trajectories $\tau_{c_k}^+$, and the negative skill $\pi_{c_k}^-$ generates negative trajectories $\tau_{c_k}^-$. The top graphs show the relationship between $c$, $\pi_c$ and $\tau_c$.

bedding should encompass skill-related information, guiding the policy on when to explore new skills or switch between existing ones. We propose that the context encoder $\psi$ should be trained by maximising the MI between the context embedding $c$, skills $\pi_c$, and trajectories $\tau_c$. To this end, we propose a novel MI optimisation objective, **Skill-aware Mutual Information (SaMI)**, defined as:

$$I_{\text{SaMI}}(c; \pi_c; \tau_c) = I(c; \tau_c) - I(c; \tau_c | \pi_c). \tag{2}$$

SaMI is defined according to interaction information [McGill, 1954], serving as a generalisation of MI for three variables $\{c, \pi_c, \tau_c\}$. Although we cannot evaluate $p(c, \pi_c, \tau_c)$ directly, we approximate it by Monte-Carlo sampling, using $K$ samples from $p(c, \pi_c, \tau_c)$. As illustrated in Figure 3, a context encoder $\psi$ trained with the objective of maximising $I_{\text{SaMI}}(c; \pi_c; \tau_c)$ converges more quickly, as $I_{\text{SaMI}}(c; \pi_c; \tau_c) \leq I(c; \tau_c)$ (see proof in Appendix B). By focusing more on skill-related information, $I_{SaMI}$ enables agents to autonomously discover a diverse range of skills for handling multiple tasks.

### 4.3   Skill-aware noise contrastive estimation: a tighter $K$-sample estimator

Despite InfoNCE's success as a $K$-sample estimator for approximating MI [Laskin et al., 2020, Eysenbach et al., 2022], its learning efficiency plunges due to limited numerical precision, which is called the log-$K$ curse, i.e., $I_{\text{InfoNCE}} \leq \log K \leq I_{\text{SaMI}}$ [Chen et al., 2021] (see proof in Appendix B). When $K \to +\infty$, we can expect $I_{\text{InfoNCE}} \approx \log K \approx I_{\text{SaMI}}$ [Guo et al., 2022]. However, increasing $K$ is too expensive, especially in complex environments with enormous negative sample space. To address this, we propose a novel $K$-sample estimator that requires a significantly smaller sample size $K \ll +\infty$. First, we define $K^*$:

**Definition 2 ($K^*$)** $K^* = |c| \cdot |\pi_c| \cdot M$ *is defined as the number of trajectories in the replay buffer (i.e., the sample space), in which $|c|$ represents the number of different context embeddings $c$, $|\pi_c|$ represents the number of different skills $\pi_c$, and $M$ is a natural number.*

To ensure that $I_{\text{InfoNCE}}$ is a tight bound of $I_{\text{SaMI}}$, we require that $I_{\text{InfoNCE}} \approx \log K \approx I_{\text{SaMI}}$ when $K \to K^*$. Under the definition of $K^*$, the replay buffer can be divided according to the different context embeddings $c$ and skills $\pi_c$ (i.e., observing context embeddings $c$ and skills $\pi_c$). In real-world robotic control tasks, the sample space size significantly increases due to multiple environmental features $e = \{e^0, e^1, ..., e^N\}$. Taking the sample space of InfoNCE as an example (Figure 4(a)), in the current task $e_1$ with context embedding $c_1$, positive samples are trajectories $\tau_{c_1}$ generated after executing the skill $\pi_{c_1}$ in task $e_1$, and negative samples are trajectories $\{\tau_{c_2}, ...\}$ from other tasks $\{e_2, ...\}$. The permutations and combinations of $N$ environmental features lead to an exponential growth in task number $|c|$, which in turn results in an increase of sample space $K^*_{\text{InfoNCE}} = |c| \cdot |\pi| \cdot M$.

We introduce a tight $K$-sample estimator, **Skill-aware Noise Contrastive Estimation (SaNCE)**, which is used to approximate $I_{\text{SaMI}}(c; \pi_c; \tau_c)$ with $K^*_{\text{SaNCE}} < K^*_{\text{InfoNCE}}$. For SaNCE, both positive samples $\tau_{c_1}^+$ and negative samples $\tau_{c_1}^-$ are sampled from the current tasks $e_1$, but are generated by executing positive skills $\pi_{c_1}^+$ and negative skills $\pi_{c_1}^-$, respectively. Here, a *positive skill* is intuitively defined by whether it is optimal for the current task $e$, with a more formal definition provided in Section 4.4. For instance, in a cube-moving task under a large friction setting, the agent executes

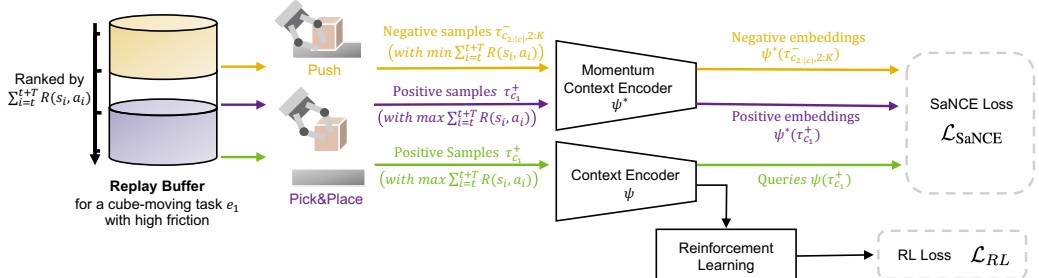

Figure 5: A practical framework for using SaNCE in the meta-training phase. During meta-training, we sample trajectories from the replay buffer for off-policy training. Queries are generated by a context encoder $\psi$, which is updated with gradients from both the SaNCE loss $\mathcal{L}_{\text{SaNCE}}$ and the RL loss $\mathcal{L}_{RL}$. Negative/Positive embeddings are encoded by a momentum context encoder $\psi^*$, which is driven by a momentum update with the encoder $\psi$. During meta-testing, the meta-trained context encoder $\psi$ embeds the current trajectory, and the RL policy takes the embedding as input together with the state for adaptation within an episode.

a skill $\pi_c^+$ after several iterations of learning, and obtains corresponding trajectories $\tau_c^+$ where the cubes leave the table surface. This indicates that the skill $\pi_c^+$ is Pick&Place and other skills $\pi_c^-$ may include Push or Flip (flipping the cube to the goal position), with corresponding trajectories $\tau_c^-$ where the cube remains stationary or rolls on the table. Formally, we can optimise the $K$-sample lower bound $I_{\text{SaNCE}}$ to approximate $I_{\text{SaMI}}$:

$$I_{\text{SaNCE}}(c; \pi_c; \tau_c | \psi, K)$$
$$= \mathbb{E}_{p(c_1, \pi_{c_1}, \tau_{c_1}^+) p(\tau_{c_1, 2:K}^-)} \left[ \log \left( \frac{K \cdot f_\psi(c_1, \pi_{c_1}, \tau_{c_1}^+)}{f_\psi(c_1, \pi_{c_1}, \tau_{c_1}^+) + \sum_{k=2}^K f_\psi(c_1, \pi_{c_1}, \tau_{c_1, k}^-)} \right) \right]$$
$$\leq I_{\text{SaMI}}(c; \pi_c; \tau_c) \tag{3}$$

where $f_\psi(c_1, \pi_{c_1}, \tau_{c_1}) = e^{\psi(\tau_{c_1})^\top \cdot \psi^*(\tau_{c_1})/\beta}$. The query $c_1 = \psi(\tau_{c_1})$ is generated by the context encoder $\psi$. For training stability, we use a momentum encoder $\psi^*$ to produce the positive and negative embeddings. SaNCE significantly reduces the required sample space size $K_{\text{SaNCE}}^*$ by sampling trajectories $\tau_c$ based on different skills $\pi_c$ (Figure 4(b)) in task $e_1$, so that $K_{\text{SaNCE}}^* = |c| \cdot |\pi_c| \cdot M = |\pi_{c_1}| \cdot M \leq K_{\text{InfoNCE}}^* (|c| = |c_1| = 1)$. Therefore, $I_{\text{SaNCE}}$ satisfies Lemma 2:

**Lemma 2** *With a context encoder $\psi$ and finite sample size $K$, we have $I_{\text{InfoNCE}}(c; \pi_c; \tau_c | \psi, K) \leq I_{\text{SaNCE}}(c; \pi_c; \tau_c | \psi, K) \leq \log K \leq I_{\text{SaMI}}(c; \pi_c; \tau_c) \leq I(c; \tau_c)$. (see proof in Appendix B)*

SaNCE can be used alone or combined with other optimisation objectives to train context encoders in Meta-RL algorithms. For instance, integrating SaNCE with InfoNCE diversifies the negative sample space, with $K_{\text{Sa+InfoNCE}}^* = \left( \sum_{i=1}^{|c|} |\pi_{c_i}^-| + |\pi_{c_1}^+| \right) \cdot M$. The sample space for $I_{\text{Sa+InfoNCE}}$ is depicted in Figure 4(c) and further analysed in detail in Appendix C.

## 4.4 Skill-aware trajectory sampling strategy

In this section, we propose a practical trajectory sampling method. Methods focusing on skill diversity often rely heavily on accurately defining and identifying individual skills [Eysenbach et al., 2018]. Some of these methods require a prior skill distribution, which is often inaccessible [Shi et al., 2022], and it is impractical to enumerate all possible skills that we hope the model to learn. Besides, we believe that distinctiveness of skills is inherently difficult to achieve — a slight difference in states can make two skills distinguishable, and not necessarily in a semantically meaningful way. Consequently, we do not directly teach any of these skills or assume any prior skill distribution. Diverse skills naturally emerge from the incentives of the SaMI learning objective in a multi-task setting, driven by the inherent need to develop generalisable skills. For example, in high-friction tasks, the agent must acquire the Pick&Place skill to avoid large frictional forces, whereas in high-mass tasks, the agent must learn the Push skill since it cannot lift the cube. In each task, we only identify whether the skills

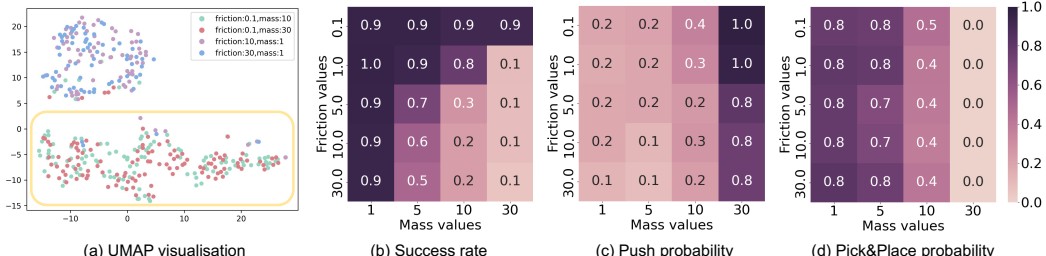

Figure 6: (a) UMAP visualisation of context embeddings for the SaCCM in the Panda-gym environment, with points in the yellow box representing the Push skill in high-mass tasks. Heatmap of (b) success rate, (c) Push skill probability, and (d) Pick&Place skill probability for SaCCM. In large-mass scenarios, the Push skill is more likely to be executed than Pick&Place.

are optimal; in this way, under a multi-task setting, the agent will acquire a set of general skills that are applicable to many tasks.

In a given task $e$, *positive skills* $\pi_c^+$ are defined as optimal skills achieving highest return $\sum_{i=t}^{t+T} R(s_i, a_i)$, whereas *negative skills* $\pi_c^-$ are those that result in lower returns. As a result, the positive sample $\tau_c^+$ consists of trajectories generated by the skills with the highest ranked returns, while the negative samples correspond to those with the lowest returns. This straightforward approach of selecting positive samples based on the ranked highest return effectively aligns with positive skills and mitigates the challenge of hard negative examples [Robinson et al., 2021]. The SaNCE loss is then minimised to bring the context embeddings of the highest return trajectories closer while distancing those of negative trajectories. By the end of training, the top-ranked trajectories in the ranked replay buffer correspond to positive samples $\tau_c^+$ with high returns, while the lower-ranked trajectories represent negative samples $\tau_c^-$ with low returns. However, at the beginning of training, it is likely that all trajectories have low returns. Therefore, our SaNCE loss is a soft variant of the $K$-sample SaNCE:

$$\mathcal{L}_{\text{SaNCE}} = -\max\left(||\psi(\tau_c^+), \psi(\tau_c^-)||_{L2}, 1\right) \cdot I_{\text{SaNCE}} \tag{4}$$

where $||\cdot||_{L2}$ represents the Euclidean distance [Tabak, 2014]. Figure 5 provides a practical framework of SaNCE, with a cube-moving example task $e_1$ under high friction. In task $e_1$, the positive skill $\pi_{c_1}^+$ is the Pick&Place skill, which is used to generate queries $\psi(\tau_{c_1}^+)$ and positive embeddings $\psi^*(\tau_{c_1}^+)$; after executing Push skill we get negative samples $\tau_{c_1}^-$ and negative embeddings $\psi^*(\tau_{c_1}^-)$.

## 5 Experiments

Our experiments aim to answer the following questions: (1) Does optimising SaMI lead to increased returns during training and zero-shot generalisation (see Table 1 and 2)?; (2) Does SaMI help the RL agents to be versatile and embody multiple skills (see Figure 6)?; (3) Can SaNCE overcome the log-$K$ curse in sample-limited scenarios (see Table 1 and 2, and Section 5.4)?

### 5.1 Experimental setup

**Modified benchmarks with multiple environmental features.**[1] We evaluate our method on two benchmarks, Panda-gym [Gallouédec et al., 2021] and MuJoCo [Todorov et al., 2012] (details in Sections 5.2 and 5.3). The benchmarks are modified to be influenced by multiple environmental features, which are sampled at the start of each episode during meta-training and meta-testing. During meta-training, we uniform-randomly select a combination of environmental features from a training task set. At test time, we evaluate each algorithm in unseen tasks with environmental features outside the training range. Generalisation performance is measured in two different regimes: moderate and extreme. The moderate regime draws environmental features from a closer range to the training range compared to the extreme. Our results report the mean and standard deviation of models trained over five seeds in both training and test tasks. Further details are available in Appendix D.

---

[1]Our modified benchmarks are open-sourced at https://github.com/uoe-agents/Skill-aware-Panda-gym

**Baselines.** The loss function of the context encoder in all algorithms consists of two key components: the RL loss and the contrastive loss. The RL loss $\mathcal{L}_{RL}$, which is the same across all methods, corresponds to the RL value function loss. Our primary comparison focuses on the contrastive loss, taking the form of either SaNCE, InfoNCE, or no contrastive loss. Accordingly, we select baselines based on contrastive loss: **CCM** [Fu et al., 2021], which utilises InfoNCE, and **TESAC** [Yu et al., 2020], which relies solely on the RL loss, allowing assessment of the context encoder without contrastive loss. Since CCM and TESAC use an RNN encoder, we also include **PEARL** [Rakelly et al., 2019], which utilises an MLP context encoder and follows the similar RL loss. Additionally, Appendix G includes comparisons with DOMINO [Mu et al., 2022] and CaDM [Lee et al., 2020], using the same environmental setup in the MuJoCo benchmark.

**Our method.**[2] We use Soft Actor-Critic (SAC) [Haarnoja et al., 2018] as the base RL algorithm, training agents for 1.6 million timesteps in each environment (details in Appendix D.3). SaNCE is a simple objective based on MI that can be used to train any context encoder. We integrate SaNCE into two Meta-RL algorithms: (1) **SaTESAC** is TESAC with SaNCE, which uses SaNCE for contrastive learning, using a $|c|$ times smaller sample space (Figure 4(b)); (2) **SaCCM** is CCM with SaNCE, where the contrastive learning combines InfoNCE and SaNCE, as shown in Figure 4(c).

### 5.2 Panda-gym

**Task description.** Our modified Panda-gym benchmark involves a robot arm control task using the Franka Emika Panda [Gallouédec et al., 2021], where the robot moves a cube to a target position. Unlike previous works, we simultaneously modify multiple environmental features (cube mass and table friction) that characterise the transition dynamics, and the robot can flexibly execute different skills (Push and Pick&Place) for different tasks. This environment requires high skill diversity; for instance, the agent must use Pick&Place in high-friction tasks and Push in high-mass tasks.

Table 1: Comparison of success rate $\pm$ standard deviation with baselines in Panda-gym (over 5 seeds). **Bold text** signifies the highest average return. $*$ next to the number means that the algorithm with SaMI has statistically significant improvement over the same algorithm without SaMI. All significance claims based on paried t-tests with significance threshold of $p < 0.05$.

|         | Training      | Test (moderate) | Test (extreme)  |
|---------|---------------|-----------------|-----------------|
| PEARL   | 0.42±0.19     | 0.10±0.06       | 0.11±0.05       |
| TESAC   | 0.50±0.22     | 0.31±0.20       | 0.22±0.21       |
| CCM     | 0.80±0.19     | 0.49±0.23       | 0.29±0.28       |
| SaTESAC | 0.92±0.04*    | 0.56±0.24*      | **0.37±0.34***  |
| SaCCM   | **0.93±0.05***| **0.57±0.26***  | 0.36±0.35*      |

**Results and skill analysis.** As shown in Table 1, SaTESAC and SaCCM achieve superior generalisation performance compared to PEARL, TESAC, and CCM, with a smaller sample space. The t-test results in Table 1 show that SaMI significantly improves success rates across training, moderate, and extreme test sets at a 0.05 significance level. Video demos[2] show that agents equipped with SaMI acquired multiple skills (Push, Pick&Place) to handle various tasks. When faced with an unknown task, the agents explore by attempting to lift the cube, infer the context, and adjust their skills accordingly within the episode. We visualised the context embeddings using UMAP [McInnes et al., 2020] (Figure 6(a) and Appendix F) and t-SNE [Van der Maaten and Hinton, 2008] (Appendix F), plotting the final step from 100 tests per task. Skills were identified through contact points between the end effector, cube, and table (see Appendix D for more details), and heatmaps [Waskom, 2021] were used to visualise executed skills. Figure 6 shows that SaCCM agents learned the Push skill for large cube masses (30 Kg, 10 Kg) and Pick&Place for smaller masses, while CCM showed no clear skill grouping (Figure 16 in Appendix F.1). Overall, SaMI incentivises agents to autonomously learn diverse skills, enhancing generalisation across a wider range of tasks. Specifically, through the cycle of effective exploration, context inference, and adaptation, diverse skills emerge solely from the data. Further visualisation results are in Appendix F.

### 5.3 MuJoCo

**Task description.** We extended the modified MuJoCo benchmark introduced in DOMINO [Mu et al., 2022] and CaDM [Lee et al., 2020]. It contains ten typical robotic control environments based on the MuJoCo physics engine [Todorov et al., 2012]. Hopper, Walker, Half-cheetah, Ant, HumanoidStandup, and SlimHumanoid are influenced by continuous environmental features (i.e., mass, damping) that affect transition dynamics. Crippled Ant, Crippled Hopper, Crippled Walker, and

---

[2]Our code, video demos and experimental data are available at https://github.com/uoe-agents/SaMI

Table 2: Comparison of average return $\pm$ standard deviation with baselines in modified MuJoCo benchmark (over 5 seeds). **Bold number** signifies the highest return. $*$ next to the number means that the algorithm with SaMI has statistically significant improvement over the same algorithm without SaMI. All significance claims based on t-tests with significance threshold of $p < 0.05$.

| | Crippled Ant | | | Crippled Half-cheetah | | |
|---|---|---|---|---|---|---|
| | Training | Test (moderate) | Test (extreme) | Training | Test (moderate) | Test (extreme) |
| PEARL | 1682±73 | 996±21 | 888±31 | 1998±973 | 698±548 | 746±1092 |
| TESAC | 2139±90 | 1952±40 | 1048±124 | 3967±955 | 874±901 | 846±849 |
| CCM | 2361±114 | 2047±83 | 1527±301 | 3481±488 | 821±575 | 873±914 |
| SaTESAC | **2638±406** | **2379±528** | **2131±132**$^*$ | 4328±1092 | **1143±664**$^*$ | **1540±1094** |
| SaCCM | 2355±170 | 2310±314 | 2007±68$^*$ | **4478±1131**$^*$ | 1007±568 | 1027±782 |

| | Ant | | | Half-cheetah | | |
|---|---|---|---|---|---|---|
| | Training | Test (moderate) | Test (extreme) | Training | Test (moderate) | Test (extreme) |
| PEARL | 5153±581 | 3873±235 | 3802±409 | 5802±773 | 2190±970 | 1346±692 |
| TESAC | 6789±451 | 4705±279 | 4108±369 | 6298±2310 | 3173±1210 | 1159±338 |
| CCM | 6901±567 | 5179±902 | 4700±696 | 6955±788 | 3963±622 | 1325±269 |
| SaTESAC | 7314±545 | 5513±648$^*$ | 4940±531$^*$ | **7430±1026** | **4058±890** | 1780±102$^*$ |
| SaCCM | **7478±539** | **5717±488** | **5215±377** | 7154±965 | 3849±689 | **1926±218**$^*$ |

| | SlimHumanoid | | | HumanoidStandup | | |
|---|---|---|---|---|---|---|
| | Training | Test (moderate) | Test (extreme) | Training | Test (moderate) | Test (extreme) |
| PEARL | 6947±3541 | 3697±2674 | 2018±907 | 95456±13445 | 63242±13546 | 64224±15467 |
| TESAC | 8437±1798 | 6989±1301 | 3760±308 | 158384±14455 | 153944±15046 | 74220±19980 |
| CCM | 7696±1907 | 5784±531 | 2887±1058 | 146480±33745 | 154601±16291 | 94991±15258 |
| SaTESAC | **10216±1620** | **7886±2203** | 6123±1403$^*$ | 178142±10081$^*$ | 168337±12123 | 133335±24607$^*$ |
| SaCCM | 9312±705 | 7430±1587 | **6473±2001**$^*$ | **187930±19338**$^*$ | **181033±14628** | **141750±27426** |

| | Hopper | | | Crippled Hopper | | |
|---|---|---|---|---|---|---|
| | Training | Test (moderate) | Test (extreme) | Training | Test (moderate) | Test (extreme) |
| PEARL | 934±242 | 874±366 | 799±298 | 3091±298 | 2387±656 | 456±235 |
| TESAC | 1492±59 | **1499±35** | **1459±72** | **3575±192** | 3298±551 | 722±161 |
| CCM | 1484±54 | 1446±64 | 1452±58 | 3455±301 | **3409±239** | 1009±289 |
| SaTESAC | **1502±20** | 1453±39 | 1447±14 | 3391±84 | 3262±166 | 1839±130 |
| SaCCM | 1462±45 | 1462±14 | 1451±67 | 3449±103 | 3390±211 | **2059±221**$^*$ |

| | Walker | | | Crippled Walker | | |
|---|---|---|---|---|---|---|
| | Training | Test (moderate) | Test (extreme) | Training | Test (moderate) | Test (extreme) |
| PEARL | 7524±2455 | 3355±2555 | 1984±356 | 7899±2532 | 4377±2563 | 2965±1426 |
| TESAC | 7747±1772 | 4355±1530 | 2581±407 | 9908±1561 | 5929±1971 | 3041±912 |
| CCM | 8136±557 | 5476±803 | 2519±682 | 10317±1137 | 6233±1869 | 3098±821 |
| SaTESAC | **8675±752** | **5840±676** | 3632±404$^*$ | 10389±1031 | **8387±1291**$^*$ | 4280±485$^*$ |
| SaCCM | 8361±586 | 5779±691 | 3481±332$^*$ | **10496±951** | 8235±1212 | **4824±839**$^*$ |

Crippled Half-cheetah are more challenging due to the addition of discrete environmental features (i.e., randomly crippled leg joints), requiring agents to master different skills (e.g., switching from running to crawling after a leg is crippled).

**Results and skill analysis.** Table 2 shows the average return of our method and baselines on training and test tasks. SaTESAC and SaCCM achieved higher returns in most tasks, except for Ant, Half-Cheetah, and Hopper, where only a single skill was needed. For instance, the Hopper robot learned to hop forward, adapting to different mass values. When environments become complex and require diverse skills for different tasks (Crippled Ant, Crippled Hopper, Crippled Half-Cheetah, SlimHumanoid, HumanoidStandup, and Crippled Walker), SaNCE brings significant improvements. For example, when the Ant robot has 3 or 4 legs available, it learns to roll to generalise across varying mass and damping. In more challenging zero-shot settings, when only 2 legs are available, the ant robot can no longer roll and it adapts by walking using its 2 healthy legs. This aligns with the results in Table 2, where SaMI significantly improved performance in extreme test sets. In summary, i) SaMI helps the RL agents to be versatile and embody multiple skills; ii) SaMI leads to increased returns

during training and zero-shot generalisation, especially in environments that require different skills. Our video demos[2] and visualisation results in Appendix F.2 show different skills in all environments.

### 5.4 Analysis of the $\log\text{-}K$ curse in sample-limited scenarios

This section analyses whether SaNCE can overcome the $\log\text{-}K$ curse. During training, environmental features are sampled at the start of each episode, requiring the context encoder to learn the context embedding distribution across multiple tasks. Since InfoNCE samples negative examples from all tasks and SaNCE samples from the current task, SaNCE's negative sample space is $|c|$ times smaller than InfoNCE's. For instance, in the SlimHumanoid environment, where both mass and damping have five values, InfoNCE's sampling space can be 25 times larger than SaNCE's. As shown in Tables 1 and 2, RL algorithms using SaNCE achieve better or comparable performance with significantly fewer negative samples ($K$) than InfoNCE. This suggests SaNCE effectively addresses the $\log\text{-}K$ curse, and the SaMI objective helps the contrastive context encoder extract critical information for downstream RL tasks.

The number of negative samples $K$ is influenced by two hyperparameters: **buffer size**, which determines the negative sample space, and **contrastive batch size**, which controls the number of samples used to train the contrastive context encoder per update. We analysed these hyperparameters further, and as shown in Figure 7, reductions in buffer and contrastive batch size do not significantly impact the average return for SaCCM and SaTESAC, which maintain state-of-the-art performance with small buffers and batch sizes. The results in Table 2 correspond to a buffer size of 100,000 and a batch size of 12. Results across all environments (refer to Appendix E.2) show that SaNCE exhibits low sensitivity to $K$, highlighting its potential to overcome the $\log\text{-}K$ curse.

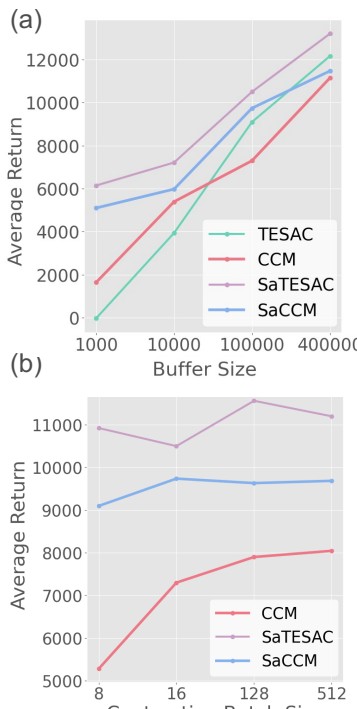

Figure 7: Effect of (a) buffer size (TESAC, CCM, SaTESAC, SaCCM) and (b) contrastive batch size (CCM, SaTESAC, SaCCM) in the SlimHumanoid environment.

## 6 Conclusion and future work

Zero-shot generalisation has been a longstanding challenge concerning Meta-RL agents, with skill diversity and sample efficiency being key to generalising to previously unseen environments. In this paper, we proposed Skill-aware Mutual Information (SaMI) to learn context embeddings for zero-shot generalisation in downstream RL tasks, and Skill-aware Noise Contrastive Estimation (SaNCE) to optimise SaMI and overcome the $\log\text{-}K$ curse, along with a practical skill-aware trajectory sampling strategy. Experimental results showed that RL algorithms equipped with SaMI achieved state-of-the-art performance in MuJoCo and Panda-gym benchmarks, particularly in zero-shot generalisation within more complex environments. During the zero-shot generalisation, when faced with an unseen task, SaMI assists agents in exploring effectively, inferring context, and rapidly adapting their skills within the current episode. SaNCE's optimisation uses a significantly smaller negative sample space than baselines, and our analysis on buffer and contrastive batch sizes demonstrated its effectiveness in addressing the $\log\text{-}K$ curse.

Given that environmental features are often interdependent, such as a cube's material correlating with friction and mass, SaMI does not introduce independence assumptions like DOMINO [Mu et al., 2022]. Therefore, future work will focus on verifying and enhancing SaMI's potential in more complex tasks where environmental features are correlated [Dunion et al., 2023a]. This will contribute to our ultimate goal: developing a generalist and versatile agent capable of working across multiple tasks and even real-world tasks in the near future.

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

# A    Proof of Lemma 1

Given a query $x$ and a set $Y = \{y_1, \ldots, y_K\}$ of $K$ random samples, containing one positive sample $y_1$ and $K - 1$ negative samples drawn from the distribution $p(y)$, a $K$-sample InfoNCE estimator is obtained by comparing pairs sampled from the joint distribution $(x, y_1) \sim p(x, y)$ with pairs $(x, y_k)$, constructed using the set of negative examples $y_{2:K}$. InfoNCE compares the positive pairs $(x, y_1)$ with the negative pairs $(x, y_k)$, where $y_k \sim y_{2:K}$, as follows:

$$I_{\text{InfoNCE}}(x; y|\psi, K) = \mathbb{E}_{p(x,y_1)p(y_{2:K})} \left[ \log \left( \frac{f_\psi(x, y_1)}{\frac{1}{K} \sum_{k=1}^{K} f_\psi(x, y_k)} \right) \right] \tag{5}$$

**Step 1.** Let us prove that the $K$-sample InfoNCE estimator is upper-bounded by $\log K$. According to Mu et al. [2022], $\frac{f_\psi(x,y_1)}{\sum_{k=1}^{K} f_\psi(x,y_k)} = \frac{f_\psi(x,y_1)}{f_\psi(x,y_1) + \sum_{k=2}^{K} f_\psi(x,y_k)} \leq 1$. So we have:

$$
\begin{aligned}
I_{\text{InfoNCE}}(x; y|\psi, K) &= \mathbb{E}_{p(x,y_1)p(y_{2:K})} \left[ \log \left( \frac{f_\psi(x, y_1)}{\frac{1}{K} \sum_{k=1}^{K} f_\psi(x, y_k)} \right) \right] \\
&= \mathbb{E}_{p(x,y)} \left[ \mathbb{E}_{p(y_{2:K})} \log \left( \frac{K \cdot f_\psi(x, y_1)}{\sum_{k=1}^{K} f_\psi(x, y_k)} \right) \right] \\
&\leq \log K
\end{aligned}
\tag{6}
$$

Hence, we have $I_{\text{InfoNCE}}(x; y|\psi, K) \leq \log K$.

**Step 2.** We have the $I(x; y) \geq I_{\text{InfoNCE}}(x; y|\psi, K)$ according to:

**Proposition 1** *[Poole et al., 2019] A $K$-sample estimator is an asymptotically tight lower bound on the MI, i.e.,*

$$I(x; y) \geq I_{\text{InfoNCE}}(x; y|\psi, K), \lim_{x \to +\infty} I_{\text{InfoNCE}}(x; y|\psi, K) \to I(x; y)$$

*Proof.* See Poole et al. [2019] for a neat proof of how the multi-sample estimator (e.g., InfoNCE) lower bounds MI.

**Step 3.** In this research, the context encoder $\psi$ in $f_\psi(x, y)$ is implemented using an RNN to approximate $\frac{p(y|x)}{p(y)}$ [Oord et al., 2019]. With a sufficiently powerful deep learning model for $\psi$ and a finite sample size $K$, such that $I(x; y) \geq \log K$, we can reasonably expect that $I_{\text{InfoNCE}} \approx \log K$ after a few training epochs. Therefore, during training, when $K \ll +\infty$, we always have $I(x; y) \geq \log K$.

*Proof.* See Chen et al. [2021] for more detailed proof.

**Step 4.** Let us prove that the $K$-sample InfoNCE bound is asymptotically tight. The specific choice of the context encoder $\psi$ influences the tightness of the $K$-sample NCE bound. InfoNCE [Oord et al., 2019] sets $f_\psi(x, y) \propto \frac{p(y|x)}{p(y)}$ to model a density ratio that preserves the MI between $x$ and $y$, where $\propto$ stands for 'proportional to' (i.e., up to a multiplicative constant). Substituting

$f_\psi(x,y) = f_\psi^*(x,y) = \frac{p(y|x)}{p(y)}$ into InfoNCE, we obtain:

$$I_{\text{InfoNCE}}(x;y|\psi,K) = \mathbb{E}\left[\log\left(\frac{f_\psi^*(x,y_1)}{\sum_{k=1}^K f_\psi^*(x,y_k)}\right)\right] + \log K$$

$$= -\mathbb{E}\left[\log\left(1 + \frac{p(y)}{p(y|x)}\sum_{k=2}^K \frac{p(y_k|x)}{p(y_k)}\right)\right] + \log K$$

$$\approx -\mathbb{E}\left[\log\left(1 + \frac{p(y)}{p(y|x)}(K-1)\mathbb{E}_{y_k\sim p(y)}\frac{p(y_k|x)}{p(y_k)}\right)\right] + \log K$$

$$= -\mathbb{E}\left[\log\left(1 + \frac{p(y_1)}{p(y_1|x)}(K-1)\right)\right] + \log K$$

$$\approx -\mathbb{E}\left[\log\frac{p(y)}{p(y|x)}\right] - \log(K-1) + \log K$$

$$= I(x;y) - \log(K-1) + \log K \tag{7}$$

Now taking $K \to +\infty$, the last two terms cancel out.

**Putting it together.** Combining $I(x;y) \geq \log K$ with Proposition 1 and Equation (6), we have Lemma 1:

$$I_{\text{InfoNCE}}(x;y|\psi,K) \leq \log K \leq I(x;y). \tag{8}$$

Moreover, according to Equation (7), as the sample size $K \to +\infty$, the $K$-sample InfoNCE bound becomes sharp and approaches the true MI $I(x;y)$, i.e., $I_{\text{InfoNCE}}(x;y|\psi,K) \approx \log K \approx I(x;y)$.

# B Proof for Lemma 2

**Step 1.** According to Lemma 1, we have $I_{\text{InfoNCE}}(c;\pi_c;\tau_c|\psi,K) \leq \log K \leq I_{\text{SaMI}}(c;\pi_c;\tau_c)$ (shown in Figure 3).

**Step 2.** Let us prove that SaNCE is a $K$-sample SaNCE estimator and is upper bounded by $\log K$. Since $\frac{f_\psi(c,\pi_c,\tau_c^+)}{f_\psi(c,\pi_c,\tau_c^+)+\sum_{k=2}^K f_\psi(c,\pi_c,\tau_{c,k}^-)} \leq 1$ [Mu et al., 2022], we have:

$$I_{\text{SaNCE}}(c;\pi_c;\tau_c|\psi,K)$$

$$= \mathbb{E}_{p(c_1,\pi_{c_1},\tau_{c_1}^+)p(\tau_{c_1,2:K}^-)}\left[\log\left(\frac{K \cdot f_\psi(c_1,\pi_{c_1},\tau_{c_1}^+)}{f_\psi(c_1,\pi_{c_1},\tau_{c_1}^+)+\sum_{k=2}^K f_\psi(c_1,\pi_{c_1},\tau_{c_1,k}^-)}\right)\right]$$

$$= \mathbb{E}_{p(c_1,\pi_{c_1})}\left[\mathbb{E}_{p(\tau_{c_1,2:K}^-)}\log\left(\frac{K \cdot f_\psi(c_1,\pi_{c_1},\tau_{c_1}^+)}{f_\psi(c_1,\pi_{c_1},\tau_{c_1}^+)+\sum_{k=2}^K f_\psi(c_1,\pi_{c_1},\tau_{c_1,k}^-)}\right)\right]$$

$$\leq \log K \tag{9}$$

Thus, we obtain $I_{\text{SaNCE}}(c;\pi_c;\tau_c|\psi,K) \leq \log K$, similar to Equation (6).

**Step 3.** With the definition of $K^*$, we can prove that $I_{\text{InfoNCE}}(c;\pi_c;\tau_c|\psi,K) \leq I_{\text{SaNCE}}(c;\pi_c;\tau_c|\psi,K)$ with the same sample size $K$. In task $e_1$, SaNCE obtains positive and negative samples from the current task $e_1$. Since the variable $c = c_1$ is constant, we have:

$$I_{\text{SaNCE}}(c;\pi_c;\tau_c|\psi,K)$$

$$= \mathbb{E}_{p(c_1,\pi_{c_1},\tau_{c_1}^+)p(\tau_{c_1,2:K}^-)}\left[\log\left(\frac{K \cdot f_\psi(c_1,\pi_{c_1},\tau_{c_1}^+)}{f_\psi(c_1,\pi_{c_1},\tau_{c_1}^+)+\sum_{k=2}^K f_\psi(c_1,\pi_{c_1},\tau_{c_1,k}^-)}\right)\right]$$

$$\leq \mathbb{E}_{p(c_1,\pi_{c_1},\tau_{c_1}^+)p(\tau_{c_1,2:K_{\text{SaNCE}}^*}^-)}\left[\log\left(\frac{K_{\text{SaNCE}}^* \cdot f_\psi(c_1,\pi_{c_1},\tau_{c_1}^+)}{f_\psi(c_1,\pi_{c_1},\tau_{c_1}^+)+\sum_{k=2}^{K_{\text{SaNCE}}^*} f_\psi(c_1,\pi_{c_1},\tau_{c_1,k}^-)}\right)\right]$$

$$= \mathbb{E}_{p(\pi_{c_1},\tau_{c_1}^+)p(\tau_{c_1,2:K_{\text{SaNCE}}^*}^-)}\left[\log\left(\frac{K_{\text{SaNCE}}^* \cdot f_\psi(\pi_{c_1},\tau_{c_1}^+)}{f_\psi(\pi_{c_1},\tau_{c_1}^+)+\sum_{k=2}^{K_{\text{SaNCE}}^*} f_\psi(\pi_{c_1},\tau_{c_1,k}^-)}\right)\right] \quad (c_1 \text{ is constant.})$$

$$\approx \log K_{\text{SaNCE}}^* \tag{10}$$

The required sample size is $K^*_{\text{SaNCE}} = |c_1| \cdot |\pi| \cdot M = |\pi| \cdot M$. As $K \to K^*_{\text{SaNCE}}$, we have $I_{\text{SaNCE}}(c; \pi_c; \tau_c|\psi, K) \approx I_{\text{SaMI}}(c; \pi_c; \tau_c)$. Correspondingly, for InfoNCE, in the current task $e_1$ with context embedding $c_1$, positive samples are trajectories $\tau_1$ generated after executing the skill $\pi_1$ in task $e_1$, while negative samples are trajectories $\{\tau^-_{c_2}, ...\}$ from other tasks $\{e_2, ...\}$. Under the definition of $K^*$, we have:

$$I_{\text{InfoNCE}}(c; \pi_c; \tau_c|\psi, K)$$

$$= \mathbb{E}_{p(c, \pi_c, \tau_{c_1})p(\tau_{c_{2:|c|}, 2:K})} \left[ \log \left( \frac{K \cdot f_\psi(c, \pi_c, \tau_{c_1})}{f_\psi(c, \pi_c, \tau_{c_1}) + \sum_{k=2}^{K} f_\psi(c, \pi_c, \tau_{c_{2:|c|}, k})} \right) \right]$$

$$\leq \mathbb{E}_{p(c, \pi_c, \tau_{c_1})p(\tau_{c_{2:|c|}, 2:K^*_{\text{InfoNCE}}})} \left[ \log \left( \frac{K^*_{\text{InfoNCE}} \cdot f_\psi(c, \pi_c, \tau_{c_1})}{f_\psi(c, \pi_c, \tau_{c_1}) + \sum_{k=2}^{K^*_{\text{InfoNCE}}} f_\psi(c, \pi_c, \tau_{c_{2:|c|}, k})} \right) \right]$$

$$\approx \log K^*_{\text{InfoNCE}}, \tag{11}$$

where $K^*_{\text{InfoNCE}} = |c| \cdot |\pi| \cdot M \approx |c| \cdot K^*_{\text{SaNCE}}$. In real-world robotic control tasks, the sample space size increases significantly due to multiple environmental features $e = \{e^0, e^1, \ldots, e^N\}$. The number of different tasks $|c|$ grows exponentially due to the permutations and combinations of the $N$ environmental features. When the current task $e_1$ has context embedding $c_1$, the $c_{2:|c|}$ refer to the context embeddings for the other tasks. As $K \to K^*_{\text{InfoNCE}}$, we have $I_{\text{InfoNCE}}(c; \pi_c; \tau_c|\psi, K) \approx I_{\text{SaMI}}(c; \pi_c; \tau_c)$. Thus, during the training process, $I_{\text{InfoNCE}}(c; \pi_c; \tau_c|\psi, K) \leq I_{\text{SaNCE}}(c; \pi_c; \tau_c|\psi, K)$ with the same sample size $K$.

**Step 4.** According to the definition of SaMI in Equation (2), we have $I_{\text{SaMI}}(c; \pi_c; \tau_c) \leq I(c; \tau_c)$, as illustrated using Venn diagrams in Figure 8 (a) and (b). SaMI is formulated based on interaction information [McGill, 1954], which aims to capture the relationships among multivariate variables by quantifying the amount of information (redundancy or synergy) shared among three variables. Interaction information has been less extensively studied, partly due to its challenging interpretation from both information theory and neuroscience perspectives, as it can be either positive or negative [Li, 2022].

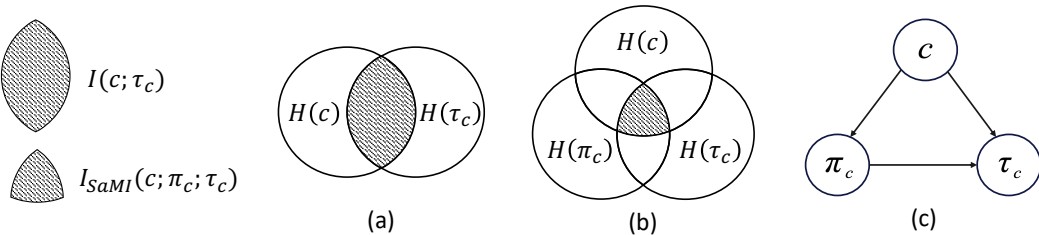

Figure 8: Venn diagrams illustrating (a) mutual information $I(c; \tau_c)$, (b) interaction information $I_{\text{SaMI}}(c; \pi_c; \tau_c)$, and (c) the MDP graph of the context embedding $c$, skill $\pi_c$, and trajectory $\tau_c$, which represents a common-cause structure [Neuberg, 2003].

In the Meta-RL setting, the MDP (causal) graph structure, shown in Figure 8 (c), illustrates that the context embedding $c$, skill $\pi_c$, and trajectory $\tau_c$ form a common-cause structure, where $c$ acts as a shared cause influencing both the skill $\pi_c$ and the trajectory $\tau_c$. Consequently, $I(\pi_c, \tau_c \mid c) < I(\pi_c, \tau_c)$. Therefore, $I_{\text{SaMI}}(c; \pi_c; \tau_c) > 0$ is guaranteed, making it interpretable in real-world robotic control tasks.

**Putting it together.** Thus, we establish Lemma 2: we always have $I_{\text{InfoNCE}}(c; \pi_c; \tau_c|\psi, K) \leq I_{\text{SaNCE}}(c; \pi_c; \tau_c|\psi, K) \leq \log K \leq I_{\text{SaMI}}(c; \pi_c; \tau_c) \leq I(c; \tau_c)$, as shown in Figure 3, while learning a skill-aware context encoder $\psi$ with the SaNCE estimator. Since $K^*_{\text{SaNCE}} \ll K^*_{\text{InfoNCE}}$, $I_{\text{SaNCE}}(c; \pi_c; \tau_c|\psi, K)$ serves as a much tighter lower bound for the true $I_{\text{SaMI}}(c; \pi_c; \tau_c)$ than $I_{\text{InfoNCE}}(c; \pi_c; \tau_c|\psi, K)$.

## C  Sample size of $I_{\text{Sa+InfoNCE}}$

In this section, we illustrate the sample size of $I_{\text{Sa+InfoNCE}}(c; \pi_c; \tau_c|\psi, K)$. Sa+InfoNCE incorporates SaNCE into InfoNCE, using positive samples $\tau^+_{c_1}$ from task $e_1$ after executing skill $\pi^+_{c_1}$, and negative

samples are trajectories $\tau^-_{c_{1:K}}$ from executing skills $\pi^-_{c_{1:K}}$ in tasks $e_{1:K}$, respectively. Therefore, this approach is equivalent to first observing the variable $c$ and then observing the variable $\pi_c$, i.e., sampling from the distribution $p(\pi_c, \tau_c | c)p(c)$. We have:

$$I_{\text{Sa+InfoNCE}}(c; \pi_c; \tau_c | \psi, K)$$

$$= \mathbb{E}_{p(c)p(\pi^+_{c_1}, \tau^+_{c_1} | c)p\left(\left(\pi^-_{2:|c|}, \tau^-_{2:|c|}\right)_{2:K}\right)} \left[ \log \left( \frac{K \cdot f_\psi(c, \pi^+_{c_1}, \tau^+_{c_1})}{f_\psi(c, \pi^+_{c_1}, \tau^+_{c_1}) + \sum_{k=2}^{K} f_\psi(c, \pi^-_{2:|c|,k}, \tau^-_{2:|c|,k})} \right) \right]$$

$$\leq \mathbb{E}_{p(c)p(\pi^+_{c_1}, \tau^+_{c_1} | c)p\left(\left(\pi^-_{2:|c|}, \tau^-_{2:|c|}\right)_{2:K^*_{\text{Sa+InfoNCE}}}\right)} \left[ \log \left( \frac{K^*_{\text{Sa+InfoNCE}} \cdot f_\psi(c, \pi^+_{c_1}, \tau^+_{c_1})}{f_\psi(c, \pi^+_{c_1}, \tau^+_{c_1}) + \sum_{k=2}^{K^*_{\text{Sa+InfoNCE}}} f_\psi(c, \pi^-_{2:|c|,k}, \tau^-_{2:|c|,k})} \right) \right]$$

$$\approx \log K^*_{\text{Sa+InfoNCE}} \tag{12}$$

It should be noted that such a combination increases the size of the negative sample space, i.e., $K^*_{\text{Sa+InfoNCE}} = \left( \sum_{i=1}^{|c|} |\pi^-_{c_i}| + |\pi^+_{c_1}| \right) \cdot M \geq K^*_{\text{SaNCE}}$. The misaligned bars in Figure 4 (c) illustrate that negative sample spaces may vary across tasks. This variation arises because we define negative samples as trajectories with low returns, making the size of the negative sample space influenced by sampling randomness. With the same number $K$ of samples, $I_{\text{Sa+InfoNCE}}$ is less precise and looser than $I_{\text{SaNCE}}$. Therefore, a trade-off between sample diversity and the precision of the $K$-sample estimator is required.

## D  Environmental setup

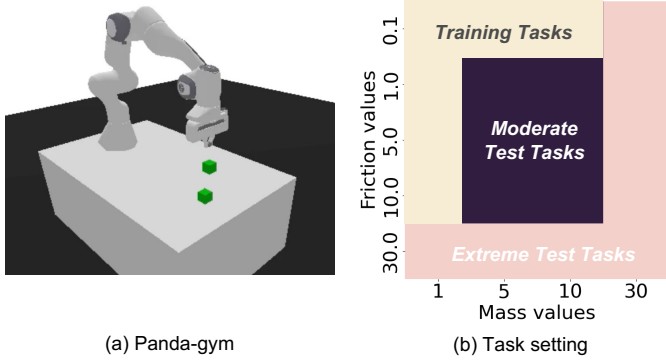

(a) Panda-gym  (b) Task setting

Figure 9: (a) Modified Panda-gym benchmarks, (b) the training tasks, moderate test tasks, and extreme test tasks. The moderate test task setting involves combinatorial interpolation, while the extreme test task setting includes unseen ranges of environmental features and represents an extrapolation.

### D.1  Modified Panda-gym

We modified the original Pick&Place task in Panda-gym [Gallouédec et al., 2021] by setting the $z$ dimension (i.e., the desired height) of the cube's goal position to $0$ [3] and maintaining the freedom of the grippers [4], allowing the agent to explore whether it should push or grasp the cube. ***Skills*** in this benchmark are defined as:

- *Pick&Place skill*: This skill specifically refers to the agent using the gripper to grasp the cube, lift it off the table, and place it in the goal position. We determine the Pick&Place skill by detecting no contact points between the table and the cube, two contact points between the robot's end effector and the cube, and the cube's height being greater than half its width.

---

[3] If $z$ is not equal to 0, the Pick&Place skill is always required to solve the tasks.

[4] In the original Push task, the grippers are blocked to ensure the agent can only push cubes. However, this restriction prevents the agent from learning Pick&Place skills, leading to "unpushable" failure in Figure 1 (b).

- *Push skill*: This skill involves the agent moving the cube on the table to the goal position, either by dragging or sliding it. We confirm the Push skill by detecting that the cube's height equals half its width.

- *Other skills*: Any behaviour modes other than Pick&Place and Push are classified as other skills.

Some elements in the RL framework are defined as follows:

***State space***: We use feature vectors that contain the cube's position (3 dimensions), cube rotation (3 dimensions), cube velocity (3 dimensions), cube angular velocity (3 dimensions), end-effector position (3 dimensions), end-effector velocity (3 dimensions), gripper width (1 dimension), desired goal (3 dimensions), and achieved goal (3 dimensions). Environmental features are not included in states.

***Action space***: The action space has 4 dimensions; the first three dimensions represent changes in the end-effector's position, and the last dimension represents the change in the gripper's width.

During training, we randomly select a combination of environmental features from a training set by sampling combinations from the following sets: mass $= 1.0$ and friction $\in \{0.1, 1.0, 5.0, 10.0\}$; mass $\in \{1.0, 5.0, 10.0\}$ and friction $= 0.1$. At test time, we evaluate each algorithm on all tasks from the moderate test setting, where mass $\in \{5.0, 10.0\}$ and friction $\in \{1.0, 5.0, 10.0\}$ (shown in Figure 9(b)), and on all tasks from the extreme test setting: mass $= 30.0$ and friction $\in \{0.1, 1.0, 5.0, 10.0, 30.0\}$; mass $\in \{1.0, 5.0, 10.0, 30.0\}$ and friction $= 30.0$ (shown in Figure 9(b)).

### D.2 Modified MuJoCo

We extended the modified MuJoCo benchmark introduced in DOMINO [Mu et al., 2022] and CaDM [Lee et al., 2020]. In our extension, four new environments were added (Walker, Crippled Hopper, Crippled Walker, HumanoidStandup), compared to the original benchmark. Additionally, in our experiments, we used a different task set design (Table 3) than those used in the DOMINO and CaDM papers. For Hopper, Walker, Half-Cheetah, Ant, HumanoidStandup, and SlimHumanoid, we used the MuJoCo physics engine environments and implemented settings from Clavera et al. [2019b] and Seo et al. [2020], scaling the mass of every rigid link by a scale factor $m$ and the damping of every joint by a scale factor $d$. For Crippled Ant, Crippled Hopper, Crippled Walker, and Crippled Half-Cheetah, we used the implementation available from Seo et al. [2020], scaled the mass of each rigid link by a factor of $m$, scaled the damping of each joint by a factor of $d$, and randomly selected joints to be uncontrollable (i.e., masking the corresponding actions with 0). Generalisation performance is measured in two different regimes: moderate and extreme, where the moderate regime draws environmental features from a range closer to the training range compared to the extreme regime. Our settings for training, extreme, and moderate test tasks are provided in Table 3.

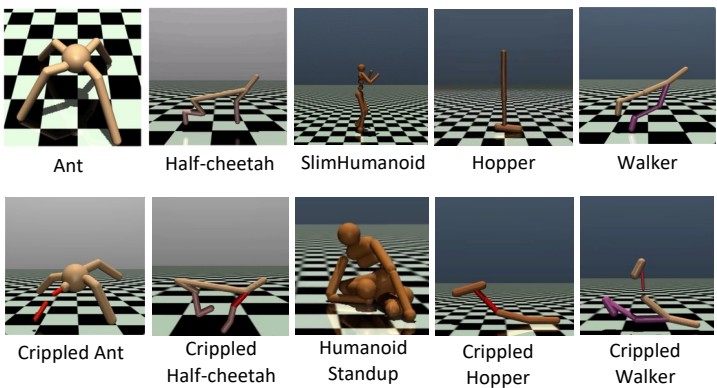

Figure 10: Ten environments in modified MuJoCo benchmark.

Table 3: Environmental features used for MuJoCo benchmark.

| | Training | Test (Moderate) | Test (Extrame) | Episode Length |
|---|---|---|---|---|
| Half-cheetah | $m \in \{0.75, 0.85, 1.0, 1.15, 1.25\}$
$d \in \{0.75, 0.85, 1.0, 1.15, 1.25\}$ | $m \in \{0.40, 0.50, 1.50, 1.60\}$
$d \in \{0.40, 0.50, 1.50, 1.60\}$ | $m \in \{0.20, 0.40, 1.60, 1.80, 4.00\}$
$d \in \{0.20, 0.40, 1.60, 1.80, 4.00\}$ | 1000 |
| Ant | $m \in \{0.40, 0.50, 1.50, 1.60\}$
$d \in \{0.40, 0.50, 1.50, 1.60\}$ | $m \in \{0.20, 0.40, 1.60, 1.80\}$
$d \in \{0.20, 0.40, 1.60, 1.80\}$ | $m \in \{0.20, 0.40, 1.60, 1.80, 4.00\}$
$d \in \{0.20, 0.40, 1.60, 1.80, 4.00\}$ | 1000 |
| Hopper | $m \in \{0.75, 1.0, 1.25\}$
$d \in \{0.75, 1.0, 1.25\}$ | $m \in \{0.40, 0.50, 1.50, 1.60\}$
$d \in \{0.40, 0.50, 1.50, 1.60\}$ | $m \in \{0.20, 0.40, 1.60, 1.80, 4.00\}$
$d \in \{0.20, 0.40, 1.60, 1.80, 4.00\}$ | 1000 |
| Crippled Hopper | $m \in \{0.75, 1.0, 1.25\}$
$d \in \{0.75, 1.0, 1.25\}$ | $m \in \{0.40, 0.50, 1.50, 1.60\}$
$d \in \{0.40, 0.50, 1.50, 1.60\}$ | $m \in \{0.20, 0.40, 1.60, 1.80, 4.0\}$
$d \in \{0.20, 0.40, 1.60, 1.80, 4.0\}$
Crippled Legs $R_1 \in \{0, 1, 2\}$ | 1000 |
| SlimHumanoid | $m \in \{0.80, 0.90, 1.0, 1.15, 1.25\}$
$d \in \{0.80, 0.90, 1.0, 1.15, 1.25\}$ | $m \in \{0.60, 0.70, 1.50, 1.60\}$
$d \in \{0.60, 0.70, 1.50, 1.60\}$ | $m \in \{0.40, 0.50, 1.70, 1.80\}$
$d \in \{0.40, 0.50, 1.70, 1.80\}$ | 1000 |
| HumanoidStandup | $m \in \{0.80, 0.90, 1.0, 1.15, 1.25\}$
$d \in \{0.80, 0.90, 1.0, 1.15, 1.25\}$ | $m \in \{0.60, 0.70, 1.50, 1.60\}$
$d \in \{0.60, 0.70, 1.50, 1.60\}$ | $m \in \{0.40, 0.50, 1.70, 1.80, 4.00\}$
$d \in \{0.40, 0.50, 1.70, 1.80, 4.00\}$ | 1000 |
| Walker | $m \in \{0.75, 1.0, 1.25\}$
$d \in \{0.75, 1.0, 1.25\}$ | $m \in \{0.40, 0.50, 1.50, 1.60\}$
$d \in \{0.40, 0.50, 1.50, 1.60\}$ | $m \in \{0.20, 0.40, 1.60, 1.80, 4.00\}$
$d \in \{0.20, 0.40, 1.60, 1.80, 4.00\}$ | 2000 |
| Crippled Walker | $m \in \{0.75, 1.0, 1.25\}$
$d \in \{0.75, 1.0, 1.25\}$
Crippled Joints (right leg) $= \{0, 1, 2\}$ | $m \in \{0.40, 0.50, 1.50, 1.60\}$
$d \in \{0.40, 0.50, 1.50, 1.60\}$
Crippled Joints (right leg) $\in \{0, 1, 2\}$ | $m \in \{0.20, 0.40, 1.60, 1.80, 4.00\}$
$d \in \{0.20, 0.40, 1.60, 1.80, 4.00\}$
Crippled Joints (left leg) $\in \{3, 4, 5\}$ | 2000 |
| Crippled Ant | $m \in \{0.75, 0.85, 1.0, 1.15, 1.25\}$
$d \in \{0.75, 0.85, 1.0, 1.15, 1.25\}$
Crippled Legs $R_1 = \varnothing$ or $R_1 \in \{0, 1, 2, 3\}$ | $m \in \{0.40, 0.50, 1.50, 1.60\}$
$d \in \{0.40, 0.50, 1.50, 1.60\}$
Crippled Legs $R_1 \in \{0, 1, 2, 3\}$ | $m \in \{0.20, 0.40, 1.60, 1.80\}$
$d \in \{0.20, 0.40, 1.60, 1.80\}$
Crippled Legs $\{R_1, R_2\} \in \{0, 1, 2, 3, 4, 5\}$ $(R_1 \neq R_2)$ | 2000 |
| Crippled Half-cheetah | $m \in \{0.75, 0.85, 1.0, 1.15, 1.25\}$
$d \in \{0.75, 0.85, 1.0, 1.15, 1.25\}$
Crippled Joints (front leg) $R_1 \in \{3, 4, 5\}$ | $m \in \{0.40, 0.50, 1.50, 1.60\}$
$d \in \{0.40, 0.50, 1.50, 1.60\}$
Crippled Joints (back leg) $R_1 \in \{0, 1, 2\}$ | $m \in \{0.20, 0.40, 1.60, 1.80\}$
$d \in \{0.20, 0.40, 1.60, 1.80\}$
Crippled Joints $\{R_1, R_2\} \in \{0, 1, 2, 3, 4, 5\}$ $(R_1 \neq R_2)$ | 2000 |

Table 4: Hyperparameters used in the Panda-gym and MuJoCo benchmarks. Most hyperparameter values remain unchanged across tasks, except for the contrastive batch size and the SaNCE loss coefficient.

| Hyperparameter | Value |
|---|---|
| Replay buffer size | 100,000 |
| Contrastive batch size | MuJoCo 12, Panda-gym 256 |
| SaNCE loss coefficient $\alpha$ | MuJoCo 1.0, Panda-gym 0.01 |
| Context embedding dimension | 6 |
| Hidden state dimension | 128 |
| Learning rate (actor, critic and encoder) | 1e-3 |
| Training frequency (actor, critic and encoder) | 128 |
| Gradient steps | 16 |
| Momentum context encoder $\psi^*$ soft-update rate | 0.05 |
| SAC target soft-update rate | critic 0.01, actor 0.05 |
| SAC batch size | 256 |
| Discount factor | 0.99 |
| Optimizer | Adam |

## D.3 Implementation details

In this section, we provide the implementation details for SaMI. We present the pseudo-code for using SaNCE during meta-training and meta-testing in Algorithms 1 and 2. Our codebase is built on top of the publicly released implementation of Stable Baselines3 by Raffin et al. [2021] and the implementation of InfoNCE by Oord et al. [2019]. A public, open-source implementation of SaMI is available at https://github.com/uoe-agents/SaMI.

**Base algorithm.** We use SAC [Haarnoja et al., 2018] for the downstream evaluation of the learned context embedding. SAC is an off-policy actor-critic method that leverages the maximum entropy framework for soft policy iteration. At each iteration, SAC performs soft policy evaluation and improvement steps. We use the same SAC implementation across all baselines and other methods. During the meta-training phase, we trained agents for 1.6 million timesteps in each environment on the Panda-gym and MuJoCo benchmarks. For meta-testing, we evaluated 100 episodes in each environment, with tasks randomly sampled from the moderate and extreme task sets.

**Encoder architecture.** In our method, the context encoder $\psi$ is modelled as a Long Short-Term Memory (LSTM) network that produces a 128-dimensional hidden state vector, which is subsequently processed through a single-layer feed-forward network to generate a 6-dimensional context embedding. We aim for the agent to complete the three steps of "explore effectively, infer, adapt" within an episode. Therefore, we initialise the hidden state and cell state of the LSTM to zero at the start of each episode. Both the actor and critic use the same context encoder to embed trajectories. For contrastive learning, SaNCE utilises a momentum encoder $\psi^*$ to generate positive and negative context embeddings [Laskin et al., 2020, He et al., 2020]. Formally, denoting the parameters of $\psi$ as $\theta_\psi$ and those of $\psi^*$ as $\theta_{\psi^*}$, we update $\theta_{\psi^*}$ as follows:

$$\theta_{\psi^*} \leftarrow m \cdot \theta_\psi + (1 - m) \cdot \theta_{\psi^*}. \tag{13}$$

Here $m \in [0, 1)$ is a soft-update rate. Only the parameters $\theta_\psi$ are updated by back-propagation. The momentum update in Equation (13) makes $\theta_{\psi^*}$ evolve more smoothly by having them slowly track the $\theta_\psi$ with $m \ll 1$ (e.g., $m = 0.05$ in this research). This means that the target values are constrained to change slowly, greatly improving the stability of learning.

**Hyperparameters.** A full list of hyperparameters is displayed in Table 4.

**Hardware.** For each experiment run we use a single NVIDIA Volta V100 GPU with 32GB memory and a single CPU.

---

**Algorithm 1** SaNCE Meta-training

---

**Require:** Batch of training tasks $\{e_n\}_{n=1,...,N}$ from $\xi_{train}(e)$, soft-update rate $m$;
  1: Initialize RL replay buffer $\mathcal{B}_{RL}$, encoder replay buffer $\mathcal{B}_{enc}$;
  2: Initialize parameters $\psi$ for context encoder, $\psi^*$ for momentum context encoder and $\phi$ for the off-policy SAC;
  3: **while** not done **do**
  4:     **for** each task $e_n$ **do**
  5:         **for** Roll-out time steps **do**
  6:             **for** time step $t <$ maximum episode length $T$ **do**
  7:                 Update context embedding $c_n \sim \psi(c_n|\tau_{c_n,0:t})$
  8:                 Roll-out policy $\pi_{c_n}(a_t|s_t, c_n)$ and accumulate transition $(s_t, a_t, r_t, s_{t+1})$;
  9:             **end for**
10:             Add trajectory $\tau_{c_n} = \{s_0, a_0, r_0, s_1, r_1, ..., s_T, a_T, r_T\}$ to replay buffer $\mathcal{B}_{RL}^n$ and $\mathcal{B}_{enc}^n$;
11:         **end for**
12:     **end for**
13:     **for** each training step **do**
14:         **for** each task $e_n$ **do**
15:             Sample RL batch $\{\tau_{c_n}\} \sim \mathcal{B}_{RL}^n$;
16:             Sample a positive sample $\tau_{c_n}^+$ for generating query with highest return, positive samples $\{\tau_{c_n}^-\}$ and negative samples $\{\tau_{c_n}^-\}$ for encoding positive and negative embeddings;
17:             Update $\phi$ with RL loss $\mathcal{L}_{RL}$;
18:             Update $\psi$ with SaNCE loss $\mathcal{L}_{SaNCE}$ and RL loss $\mathcal{L}_{RL}$;
19:             $\theta_{\psi^*} \leftarrow m \cdot \theta_\psi + (1-m) \cdot \theta_{\psi^*}$;
20:         **end for**
21:     **end for**
22: **end while**

---

**Algorithm 2** SaNCE Meta-testing

---

**Require:** Batch of training tasks $\{e_n\}_{n=1,...,N}$ from $\xi_{test}(e)$;
  1: **while** not done **do**
  2:     **for** each task $e_n$ **do**
  3:         **for** each episode **do**
  4:             **for** time step $t <$ maximum episode length $T$ **do**
  5:                 Update context embedding $c_n \sim \psi(c_n|\tau_{c_n,0:t})$
  6:                 Roll-out policy $\pi_{c_n}(a_t|s_t, c_n)$ and accumulate transition $(s_t, a_t, r_t, s_{t+1})$;
  7:             **end for**
  8:         **end for**
  9:     **end for**
10: **end while**

---

# E Additional results

## E.1 Balance contrastive and RL updates: loss coefficient $\alpha$

While past work has optimised hyperparameters to balance the contrastive loss coefficient $\alpha$ relative to the RL objective [Jaderberg et al., 2016, Bachman et al., 2019], we use both the contrastive and RL objectives with equal weight, setting $\alpha = 1.0$ for the MuJoCo benchmark and $\alpha = 0.01$ for the Panda-gym benchmark. Additionally, we analyse the effect of the loss coefficient $\alpha$ for CCM, SaTESAC, and SaCCM in the MuJoCo (Figure 12) and Panda-gym (Figure 11) benchmarks.

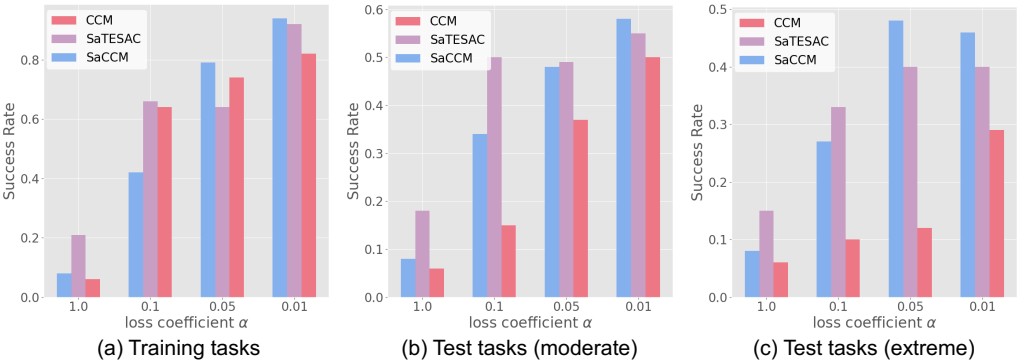

Figure 11: Loss coefficient $\alpha$ analysis of Panda-gym benchmark in training and test (moderate and extreme) tasks.

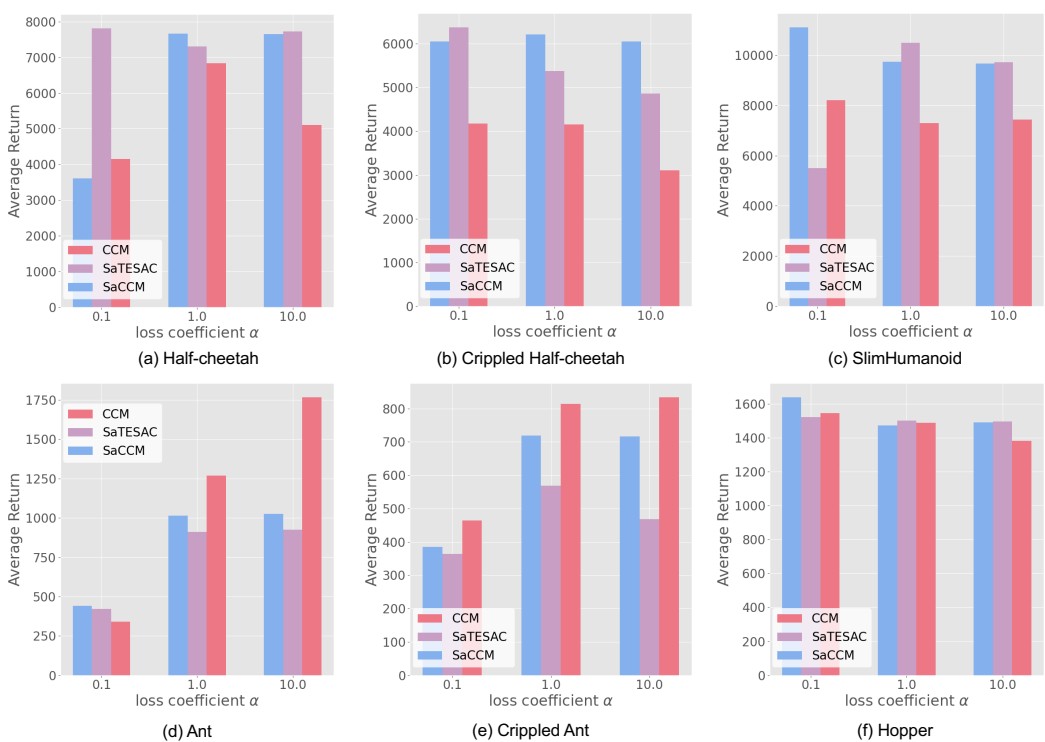

Figure 12: Loss coefficient $\alpha$ analysis of MuJoCo benchmark in training tasks.

## E.2  Result of $\log$-$K$ curse analysis

### E.2.1  Buffer size

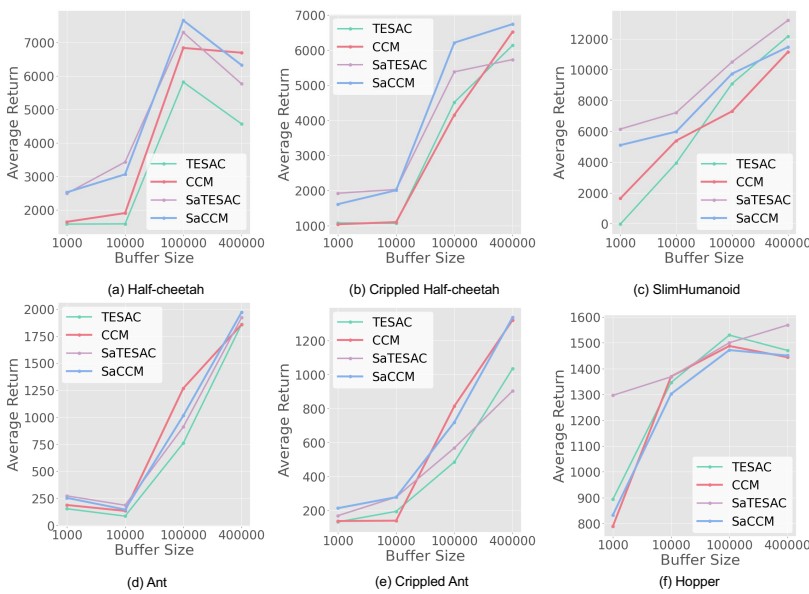

Figure 13: Comparison of different buffer sizes in the MuJoCo benchmark on training tasks (averaged over 5 seeds). Buffer sizes are 400,000, 100,000, 10,000, and 1,000.

### E.2.2  Contrastive batch size

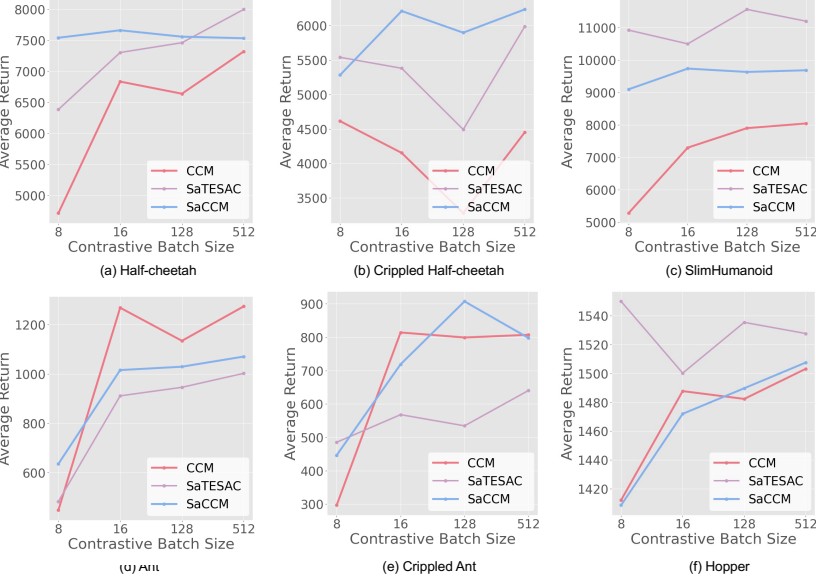

Figure 14: Comparison of different contrastive batch sizes in the MuJoCo benchmark on training tasks (averaged over 5 seeds). Contrastive batch sizes are 512, 128, 16, and 8.

# F Further skill analysis

## F.1 Panda-gym

### F.1.1 Visualisation of context embedding

We visualise the context embedding using UMAP [McInnes et al., 2020] (Figure **??**) and t-SNE [Van der Maaten and Hinton, 2008] (Figure 16). When the mass of the cube is high (30 kg and 10 kg), the agent learned the Push skill (indicated by the yellow bounding box in Figure 1(a)), whereas with lower masses, the agent learned the Pick&Place skill. However, as shown in Figure 16(b), CCM did not display clear skill grouping. This indicates that SaMI extracts high-quality skill-related information from the trajectories and enables agents to autonomously discover a diverse range of skills for handling multiple tasks.

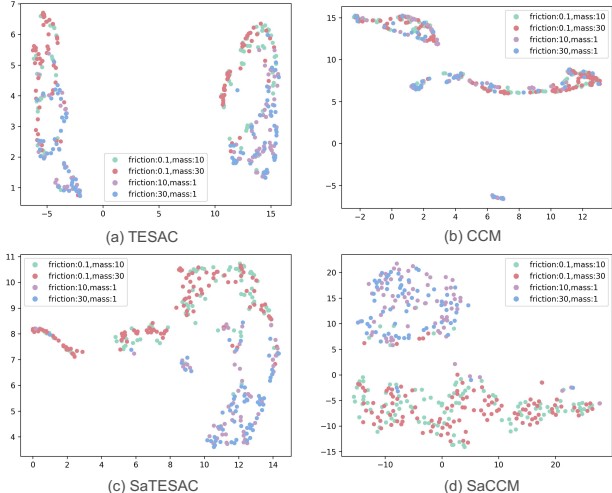

Figure 15: UMAP visualisation of context embeddings extracted from trajectories collected in the Panda-gym environments.

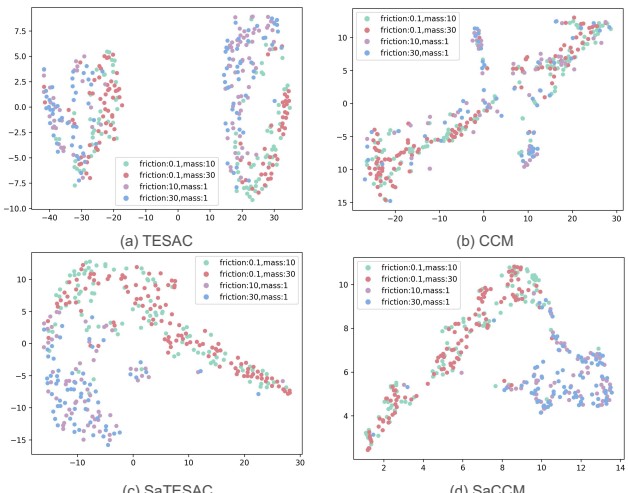

Figure 16: t-SNE visualisation of context embeddings extracted from trajectories collected in the Panda-gym environments.

### F.1.2 Heatmap of Panda-gym benchmark

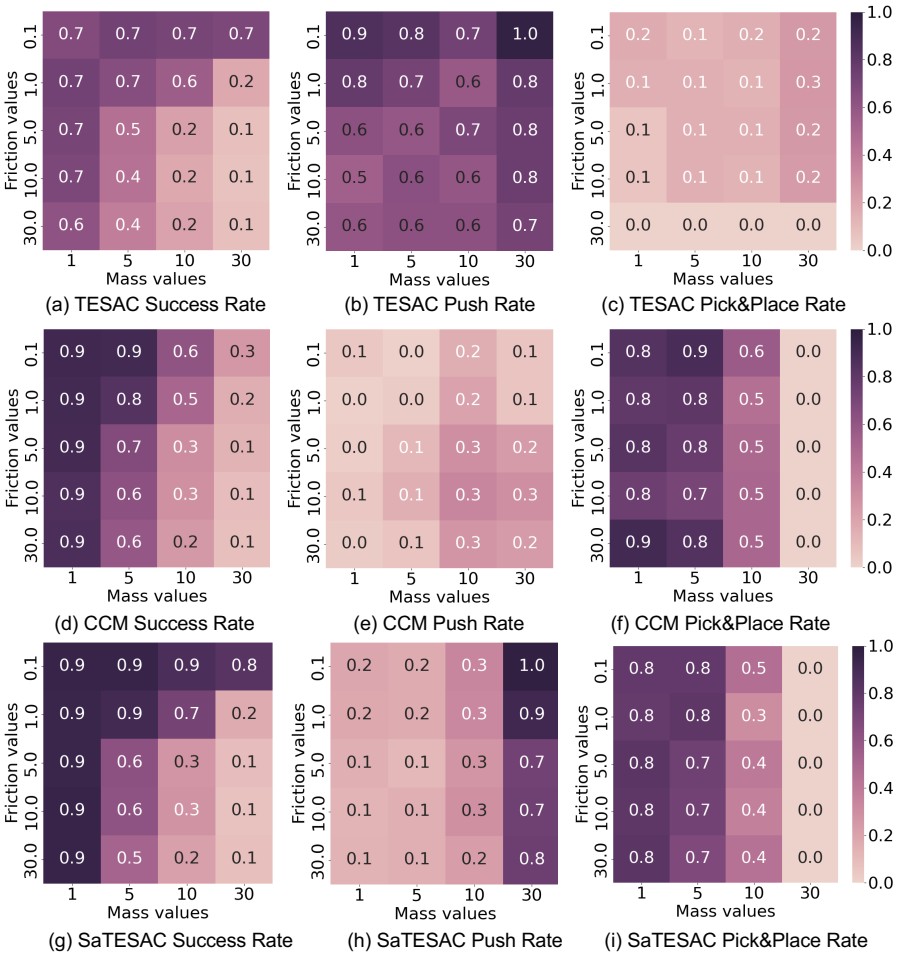

Figure 17: Heatmap of success rates and learned skills of SaCCM. For each grid, the $(i, j)^{\text{th}}$ location shows the probability of the skills executed over 100 evaluations with $(\text{mass} = i, \text{friction} = j)$.

This section presents the heatmap results and further analysis of TESAC, CCM, and SaTESAC on the Panda-gym benchmark. From the heatmap results of SaTESAC and SaCCM (Figure 6), we observe that, with higher cube masses (30 and 10 kg), the agent executed the Push skill (indicated by the clustered points within the yellow bounding box in Figure 1(a)). At lower masses, the agent executed the Pick&Place skill.

In contrast, as shown in Figures 17(e-f), CCM predominantly learned the Pick&Place skill, resulting in a drop in success rates for tasks with $mass = 30$, as the agent could not lift the cube off the table using the Pick&Place skill, as depicted in Figure 17(d). The visualisation of the context embedding (Figure 16) did not reveal clear skill grouping across different tasks.

Finally, TESAC primarily mastered the Push skill. The Push skill is relatively simpler to learn than the Pick&Place skill, as it does not require the agent to manipulate its fingers to pick up cubes. Consequently, TESAC's success rate notably decreased in environments with higher friction.

## F.2 MuJoCo

SaMI enables RL agents to be versatile and embody multiple skills. Additionally, video demos (available at https://github.com/uoe-agents/SaMI) provide a better demonstration of the different skills.

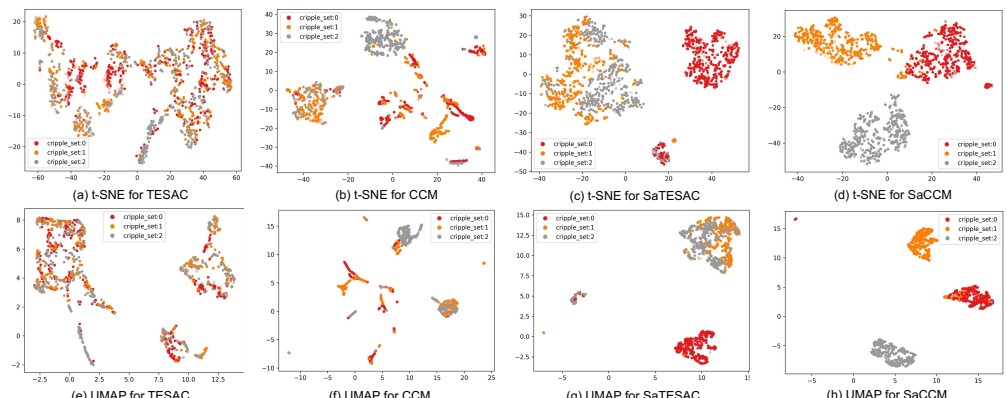

Figure 18: t-SNE and UMAP visualisations of context embeddings extracted from trajectories collected in the Crippled Half-Cheetah environment. "cripple_set" refers to the index of the crippled joint. The figure shows the context embeddings of tasks in three moderate test settings. Combined with the video demos[2] for skill analysis, the Crippled Half-Cheetah robot executed three distinct forward running skills.

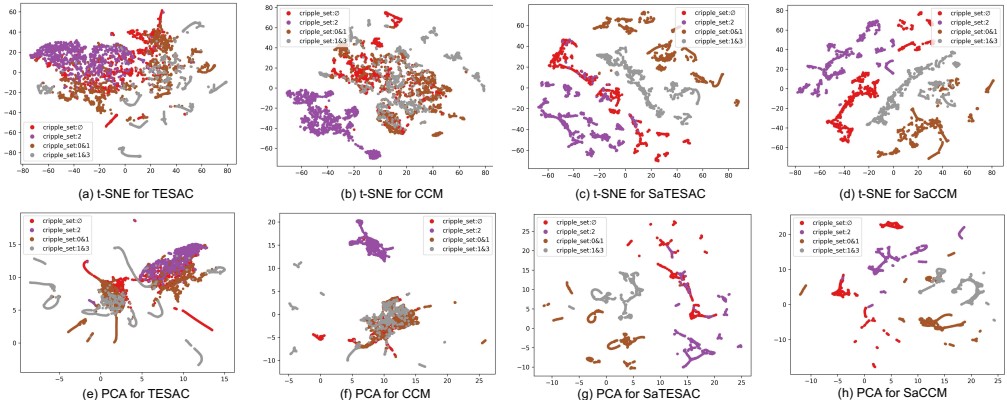

Figure 19: t-SNE and UMAP visualisations of context embeddings extracted from trajectories collected in the Crippled Ant environment. "cripple_set" refers to the index of the crippled joint. When 3 or 4 legs are available, the Ant robot (trained with SaCCM) rolls to adapt to varying mass and damping. However, with only 2 adjacent legs during zero-shot generalisation, it switches to walking. If 2 opposite legs are available, the Ant can still roll but eventually tips over. Please refer to the video demos[2].

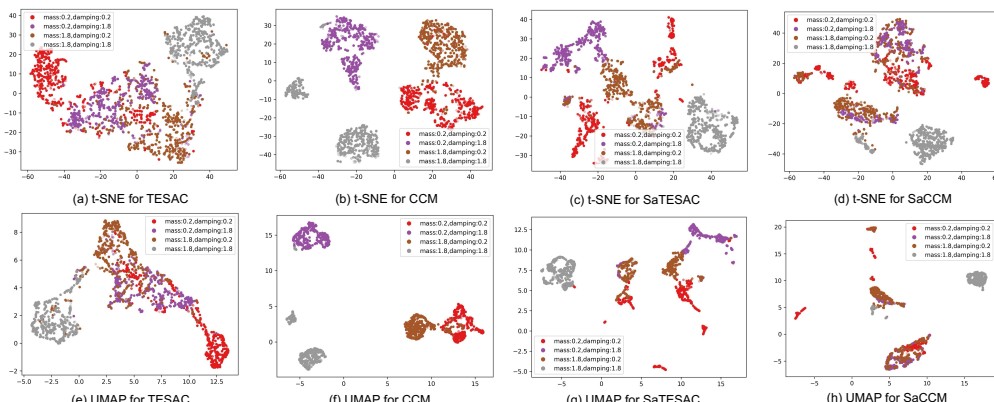

Figure 20: t-SNE and UMAP visualisations of context embeddings extracted from trajectories collected in the Half-Cheetah environment. During zero-shot generalisation, SaCCM demonstrates different skills for running forwards at various speeds, as well as skills for doing flips and faceplanting.

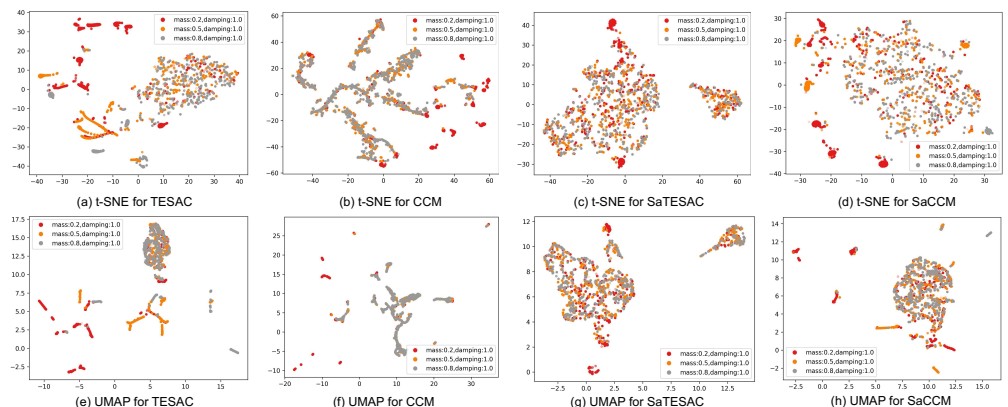

Figure 21: t-SNE and UMAP visualisations of context embeddings extracted from trajectories collected in Ant environment. The Ant robot learned a single skill, rolling, to adapt to different mass and damping values.

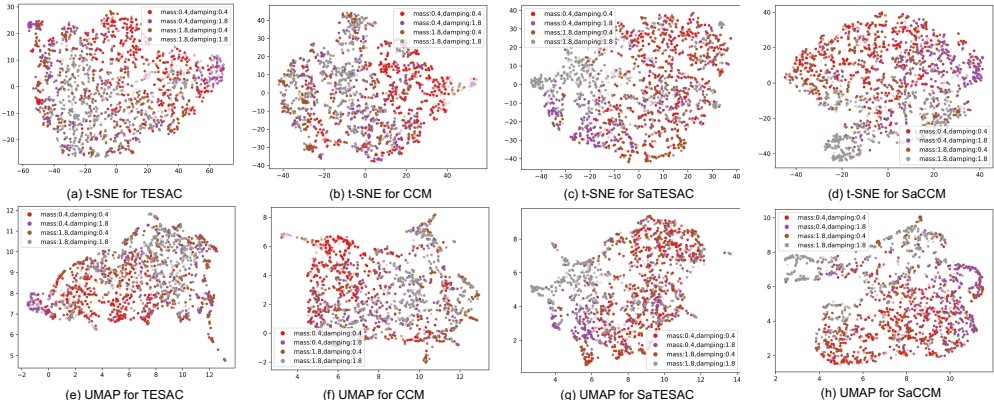

Figure 22: t-SNE and UMAP visualisations of context embeddings extracted from trajectories collected in SlimHumanoid environment. The Humanoid Robot crawls on the ground using one elbow. When the damping is relatively high (damping=1.8), the Humanoid Robot can crawl forward stably, but when the damping is low (damping=0.4), it tends to roll.

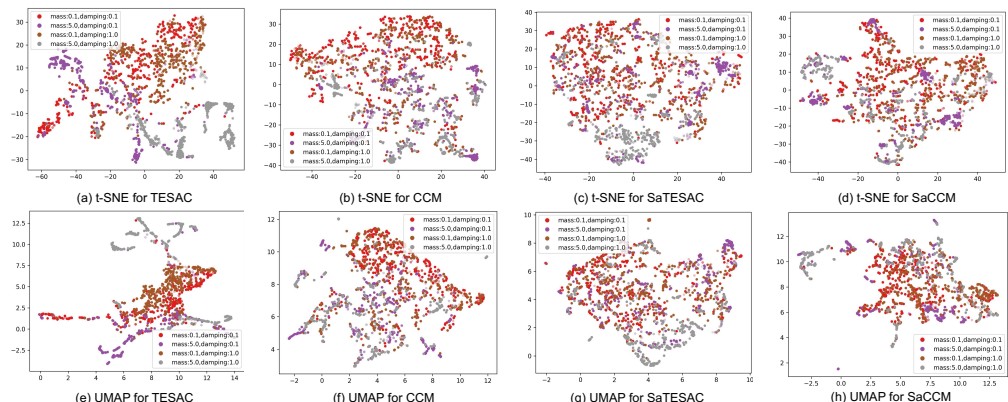

Figure 23: t-SNE and UMAP visualisations of context embeddings extracted from trajectories collected in HumanoidStandup environment. SaCCM and SaTESAC learned a sitting posture that makes it easier to stand up, allowing it to generalise well when mass and damping change.

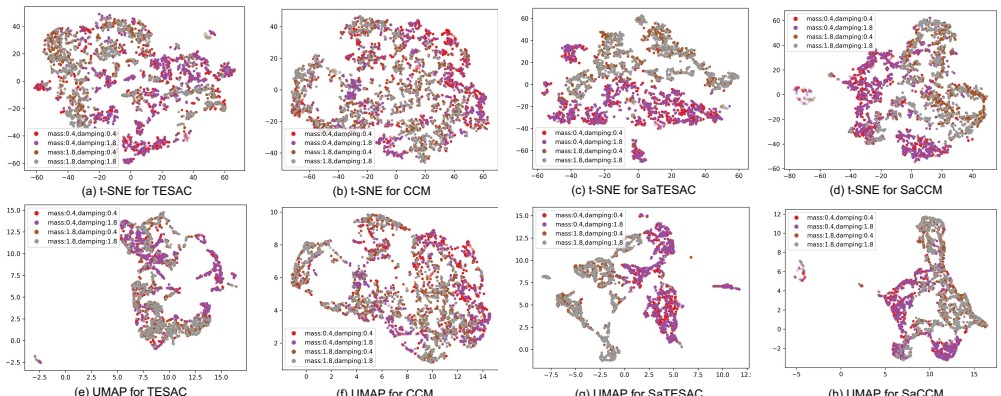

Figure 24: t-SNE and UMAP visualisations of context embeddings extracted from trajectories collected in Hopper environment. Combined with the video demos [2] for skill analysis, the plots for SaCCM and SaTESAC show two skills: 1) when the mass is low, the Hopper hops in an upright posture; 2) when the mass is higher, the Hopper hops forward on the floor.

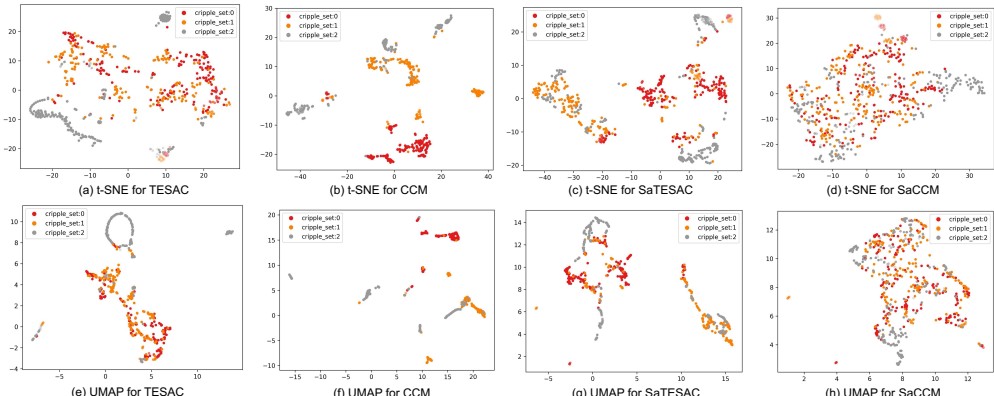

Figure 25: t-SNE and UMAP visualisations of context embeddings extracted from trajectories collected in Crippled Hopper environment. "cripple_set" refers to the index of the crippled joint. SaCCM and SaTESAC learned to take a big hop forward at the beginning (i.e., effective exploration) and then switch to different skills based on environmental feedback.

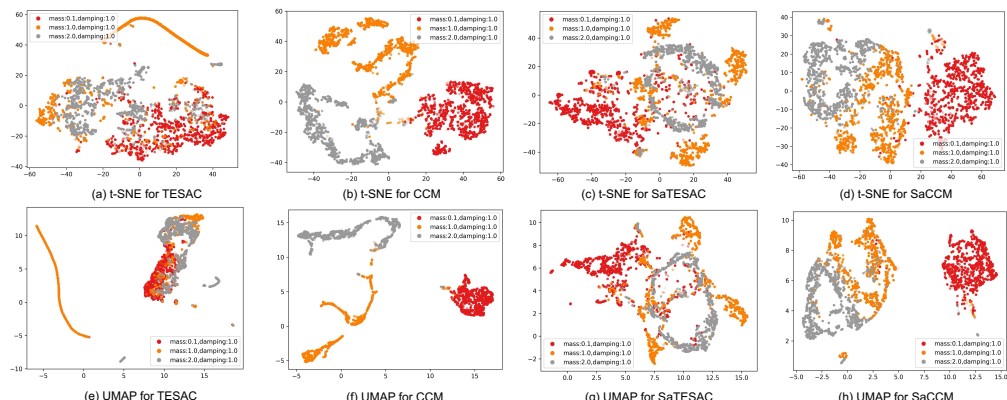

Figure 26: t-SNE and UMAP visualisations of context embeddings extracted from trajectories collected in Walker environment. The Walker learned a single skill, hopping forward on the floor through both right and left legs.

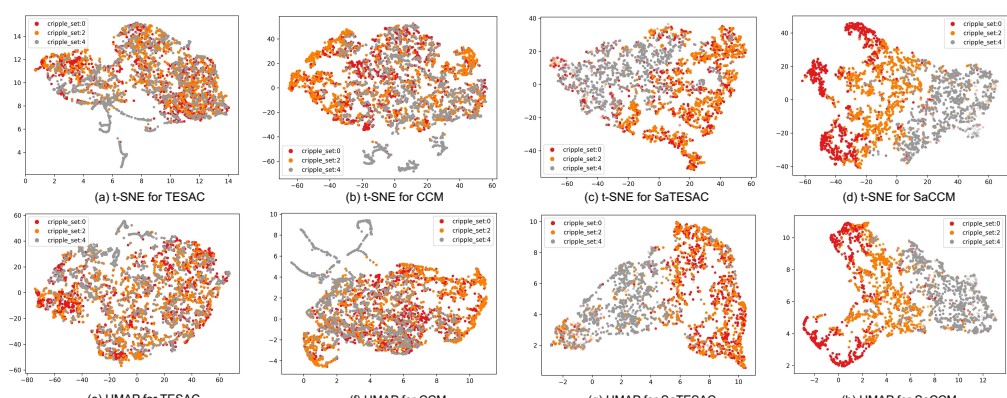

Figure 27: t-SNE and UMAP visualisations of context embeddings extracted from trajectories collected in Crippled Walker environment. "cripple_set" refers to the index of the crippled joint. The Crippled Walker (trained with SaTESAC and SaCCM) learned to hop forward using the right leg (cripple_set:0 and cripple_set:2) in the training and moderate test tasks and switched to hopping forward using the left leg (cripple_set:4) in the extreme test tasks.

Table 5: Environmental features used for MuJoCo benchmark from DOMINO and CaDM.

| | Training | Test (Moderate) | Test (Extrame) | Episode Length |
|---|---|---|---|---|
| Half-cheetah | $m \in \{0.75, 0.85, 1.0, 1.15, 1.25\}$ 
 $d \in \{0.75, 0.85, 1.0, 1.15, 1.25\}$ | $m \in \{0.40, 0.50, 1.50, 1.60\}$ 
 $d \in \{0.40, 0.50, 1.50, 1.60\}$ | $m \in \{0.20, 0.40, 1.60, 1.80\}$ 
 $d \in \{0.20, 0.40, 1.60, 1.80\}$ | 1000 |
| Ant | $m \in \{0.75, 0.85, 1.0, 1.15, 1.25\}$ 
 $d \in \{0.75, 0.85, 1.0, 1.15, 1.25\}$ | $m \in \{0.40, 0.50, 1.50, 1.60\}$ 
 $d \in \{0.40, 0.50, 1.50, 1.60\}$ | $m \in \{0.20, 0.40, 1.60, 1.80\}$ 
 $d \in \{0.20, 0.40, 1.60, 1.80\}$ | 1000 |
| SlimHumanoid | $m \in \{0.80, 0.90, 1.0, 1.15, 1.25\}$ 
 $d \in \{0.80, 0.90, 1.0, 1.15, 1.25\}$ | $m \in \{0.60, 0.70, 1.50, 1.60\}$ 
 $d \in \{0.60, 0.70, 1.50, 1.60\}$ | $m \in \{0.40, 0.50, 1.70, 1.80\}$ 
 $d \in \{0.40, 0.50, 1.70, 1.80\}$ | 500 |
| Crippled Half-cheetah | $m \in \{0.75, 0.85, 1.0, 1.15, 1.25\}$ 
 $d \in \{0.75, 0.85, 1.0, 1.15, 1.25\}$ 
 Crippled Joints: $\{0, 1, 2, 3\}$ | $m \in \{0.40, 0.50, 1.50, 1.60\}$ 
 $d \in \{0.40, 0.50, 1.50, 1.60\}$ 
 Crippled Joints: $\{4, 5\}$ | $m \in \{0.20, 0.40, 1.60, 1.80\}$ 
 $d \in \{0.20, 0.40, 1.60, 1.80\}$ 
 Crippled Joints: $\{4, 5\}$ | 1000 |

Table 6: Comparison of average return $\pm$ standard deviation with baselines in MuJoCo benchmark (over 5 seeds). The **bold text** signifies the highest average return. The numerical results for PPO+DOMINO are copied from Mu et al. [2022]; the numerical results for PPO+CaDM, Vanilla+CaDM, and PE-TS+CaDM are copied from Lee et al. [2020].

| | Ant | | | Half-cheetah | | |
|---|---|---|---|---|---|---|
| | Training | Test (moderate) | Test (extreme) | Training | Test (moderate) | Test (extreme) |
| PPO+DOMINO | 227±86 | 216±52 | | 2472±803 | 1034±476 | |
| PPO+CaDM | 268.6±77.0 | 228.8±48.4 | 199.2±52.1 | 2652.0±1133.6 | 1224.2±630.0 | 1021.1±676.6 |
| Vanilla+CaDM | 1851.0±113.7 | 1315.7±45.5 | 821.4±113.5 | 3536.5±641.7 | 1556.1±260.6 | 1264.5±228.7 |
| PE-TS+CaDM | **2848.4±61.9** | **2121.0±60.4** | **1200.7±21.8** | **8264.0±1374.0** | **7087.2±1495.6** | **4661.8±783.9** |
| SaTESAC | 908±65 | 640±117 | 532±88 | 7430±1026 | 4058±890 | 1780±102 |
| SaCCM | 928±141 | 635±94 | 555±88 | 7154±965 | 3849±689 | 1926±218 |

| | SlimHumanoid | | | Crippled Half-Cheetah | | |
|---|---|---|---|---|---|---|
| | Training | Test (moderate) | Test (extreme) | Training | Test (moderate) | Test (extreme) |
| PPO+DOMINO | 7825±1256 | 5258±1039 | | 2503±658 | 1326±491 | |
| PPO+CaDM | **10455.0±1004.9** | 4975.7±1305.7 | 3015.1±1508.3 | 2356.6±624.3 | 1454.0±462.6 | 1025.0±296.2 |
| Vanilla+CaDM | 1758.2±459.1 | 1228.9±374.0 | 1487.9±339.0 | 2435.1±880.4 | 1375.3±290.6 | 966.9±89.4 |
| PE-TS+CaDM | 1371.9±400.0 | 903.7±343.9 | 814.5±274.8 | 3294.9±733.9 | 2618.7±647.1 | 1294.2±214.9 |
| SaTESAC | 10216±1620 | **7886±2203** | 6123±1403 | 5169±730 | 2184±592 | 1628±281 |
| SaCCM | 9312±705 | 7430±1587 | **6473±2001** | **5709±744** | **2795±446** | **2115±466** |

# G   A comparison with DOMINO and CaDM

In this section, we give a brief comparison between our methods and methods from DOMINO [Mu et al., 2022] and CaDM [Lee et al., 2020] in the MuJoCo benchmark because we are using the exact same environmental setting (shown in Table 5).

DOMINO [Mu et al., 2022] is based on the InfoNCE $K$-sample estimator. Their implementation, PPO+DOMINO, is a model-free RL algorithm with a pre-trained context encoder. This encoder reduces the demand for large contrastive batch sizes during training by decoupling representation learning for each modality, simplifying tasks while leveraging shared information. However, a pre-trained encoder necessitates a large sample volume, with DOMINO training PPO agents for 5 million timesteps on the MuJoCo benchmark. In contrast, SaTESAC and SaCCM, trained for 1.6 million timesteps without pre-trained encoders, achieve considerably higher average returns across four environments (Table 6). Therefore, it is crucial to focus on extracting MI in contrastive learning that directly optimises downstream tasks, integrating rather than segregating representation learning from task performance.

CaDM [Lee et al., 2020] proposes a context-aware dynamics model adaptable to changes in dynamics. Specifically, they utilise contrastive learning to learn context embeddings, and then predict the next state conditioned on them. We copy the numerical results of PPO+CaDM, Vanilla+CaDM, and PE-TS+CaDM from CaDM [Lee et al., 2020] as their environmental setting is identical to ours, where PPO+CaDM is a model-free RL algorithm, while Vanilla+CaDM and PE-TS+CaDM are model-based. The model-free RL approach, PPO+CaDM, is trained for 5 million timesteps on the MuJoCo benchmark. As shown in Table 6, SaTESAC and SaCCM significantly outperform PPO+CaDM. The model-based RL algorithms, Vanilla+CaDM and PE-TS+CaDM, require 2 million timesteps for learning in model-based setups, compared to our fewer samples (i.e., million timesteps). In the Ant environment, Vanilla+CaDM and PE-TS+CaDM achieve higher returns than SaTESAC and SaCCM; similarly, in the Half-cheetah environment, PE-TS+CaDM outperforms them. Results in the SlimHumanoid and Crippled Half-cheetah environments show that skill-aware context embeddings are notably effective. An insight here is that our method outperforms the model-free CaCM approach, but not the model-based one. This is consistent with what is empirically found in CaDM [Lee et al., 2020]: prediction models are more effective when the transition function changes across tasks. Therefore, we consider that a model-based approach to SaMI could be an interesting extension for future work.

Table 7: The p-value of the statistical hypothesis tests (paried t-tests) for comparing the effectiveness of SaMI in MuJoCo benchmark (over 5 seeds). ∗ next to the number means that the algorithm with SaMI has statistically significant improvement over the same algorithm without SaMI at a significance level of 0.05. The "SaTESAC-TESAC" row indicates the p-value for the return improvement brought by SaMI to TESAC; the "SaCCM-CCM" row indicates the p-value for the return improvement brought by SaMI to CCM.

|  | Crippled Ant | | | Crippled Half-cheetah | | |
|---|---|---|---|---|---|---|
|  | Training | Test (moderate) | Test (extreme) | Training | Test (moderate) | Test (extreme) |
| SaTESAC-TESAC | 0.121 | 0.109 | 9.54E-07* | 0.154 | 0.0024* | 0.0889 |
| SaCCM-CCM | 0.913 | 0.108 | 0.008* | 0.04* | 0.307 | 0.106 |
|  | Ant | | | Half-cheetah | | |
|  | Training | Test (moderate) | Test (extreme) | Training | Test (moderate) | Test (extreme) |
| SaTESAC-TESAC | 0.136 | 0.034* | 0.021* | 0.346 | 0.224 | 0.004* |
| SaCCM-CCM | 0.138 | 0.275 | 0.163 | 0.73 | 0.791 | 0.005* |
|  | SlimHumanoid | | | HumanoidStandup | | |
|  | Training | Test (moderate) | Test (extreme) | Training | Test (moderate) | Test (extreme) |
| SaTESAC-TESAC | 0.139 | 0.456 | 0.006* | 0.037* | 0.129 | 0.003* |
| SaCCM-CCM | 0.113 | 0.059 | 0.008* | 0.048 | 0.027* | 0.01 |
|  | Hopper | | | Crippled Hopper | | |
|  | Training | Test (moderate) | Test (extreme) | Training | Test (moderate) | Test (extreme) |
| SaTESAC-TESAC | 0.747 | 0.089 | 0.707 | 0.459 | 0.69 | 0.088 |
| SaCCM-CCM | 0.52 | 0.599 | 0.969 | 0.967 | 0.897 | 0.0002* |
|  | Walker | | | Crippled Walker | | |
|  | Training | Test (moderate) | Test (extreme) | Training | Test (moderate) | Test (extreme) |
| SaTESAC-TESAC | 0.312 | 0.082 | 0.003* | 0.223 | 0.048* | 0.028* |
| SaCCM-CCM | 0.55 | 0.541 | 0.022* | 0.794 | 0.079 | 0.011* |

Table 8: The p-value of the statistical hypothesis tests (paried t-tests) for comparing the effectiveness of SaMI in Panda-gym benchmark (over 5 seeds). ∗ next to the number means that the algorithm with SaMI has statistically significant improvement over the same algorithm without SaMI at a significance level of 0.05. The "SaTESAC-TESAC" row indicates the p-value for the return improvement brought by SaMI to TESAC; the "SaCCM-CCM" row indicates the p-value for the return improvement brought by SaMI to CCM.

|  | Training | Test (moderate) | Test (extreme) |
|---|---|---|---|
| SaTESAC-TESAC | 0.000260* | 0.000160* | 0.001310* |
| SaCCM-CCM | 0.000230* | 0.002190* | 0.001390* |

# H   Statistical hypothesis tests (paried t-tests)

We used a t-test [Rice and Rice, 2007] to conduct a statistical hypothesis test to determine whether SaMI brought a statistically significant improvement. we reported the p-value of the t-test in MuJoCo (Table 7) and Panda-gym (Table 8) benchmarks. ∗ next to the number is used to indicate that the algorithm with SaMI has statistically significant improvement over the same algorithm without SaMI at a significance level of 0.05. From Table 7 and 8, SaMI brings significant improvement on the extreme test set in which the RL agent needs to execute diverse skills. The statistically significant test aligns with our results in the skill analysis (i.e., video demos, t-SNE and UMAP visualisation). In complex environments that require high skill diversity from the RL agent, the statistically significant improvement and higher returns/success rates brought by SaMI are evident.

