# OpenReview forum: "Skill-aware Mutual Information Optimisation for Zero-shot Generalisation in Reinforcement Learning"
_NeurIPS.cc/2024/Conference — NeurIPS 2024 poster_

### Official Review · Reviewer_Qb7x · 2024-06-16

**Soundness:** 3
**Presentation:** 2
**Contribution:** 3
**Rating:** 6
**Confidence:** 3

**Summary:**

Proposes a new contrastive learning objective for use in meta-RL with contextual policies. The new objective incorporates a notion of skills into the mutual information estimate. Theoretical and empirical evidence in support of the superiority of the proposed method is presented.

**Strengths:**

- The paper is clearly written.
- The proposed method is well motivated.
- The proposed benchmark is interesting.

**Weaknesses:**

- Meta-RL is presented as the problem of learning a contextual RL agent and a context generator for solving the task. However, there are many other kinds of meta-RL algorithms that do not use explicit context representations at all. For some canonical examples, see [RL^2](https://arxiv.org/abs/1611.02779) and [MAML](https://arxiv.org/abs/1703.03400). It is not important to include a thorough survey of other types of meta-RL methods, but for the sake of accuracy, the full scope of meta-RL should be mentioned. Especially important is to note that the proposed method requires a posterior sampling style meta-RL approach, which is not optimal at test-time compared to directly optimizing the meta-RL objective via, e.g., training a sequence model as the policy.
- Due to the above issue, it is inaccurate to say that SaNCE can be integrated with any Meta-RL algorithm.
- A new benchmark environment should come with thorough benchmarking. A good selection of contrastive meta-RL methods are included, but it would be more convincing if at least some RL^2 style method with suitable architecture (e.g. [transformers](https://proceedings.mlr.press/v162/melo22a.html) or [hypernetworks](https://proceedings.neurips.cc/paper_files/paper/2023/hash/c3fa3a7d50b34732c6d08f6f66380d75-Abstract-Conference.html)) was also run as a baseline. The current benchmarking doesn't try zero-shot meta-RL algorithms at all.
- The paper isn't self-contained in terms of understanding how the proposed method works during meta-training and meta-testing times. See questions for details.

**Questions:**

- For how many episodes does the agent interact with the environment at meta-training and testing times?
- How does the encoder accumulate across multiple episodes of interaction?
- Is there a way to use the proposed method for adaptation within an episode?

**Limitations:**

Limitations are not discussed.

---

> ### Author Rebuttal · Authors · 2024-08-06
>
> We are very grateful for the reviewer's professional review work on our paper. We would like to express our sincere appreciation for the reviewer's recognition of our work, especially regarding the writing and the motivation behind our work. We also greatly appreciate the acknowledgement of our theoretical and empirical evidence. Please let us know if further clarification is required.
>
> **Re Weakness (1) and (2):** Regarding the reviewer's question about whether SaNCE can be combined with any meta-RL algorithm (e.g., $\text{RL}^2$ and MAML), SaNCE is a simple objective based on mutual information that can be used to train any context encoder as defined in Section 4.2 “Skill-aware noise contrastive estimation: a tighter $K$-sample estimator” in our paper and enable contextual RL agents to autonomously discover skills. We have now specified the types of algorithms where SaNCE can be used to make the scope clearer in the new manuscript, specifically in Section 5.1 “Experimental Setup” under the “Our Methods” paragraph.
>
> **Re Weakness (3):** Please note that the modified MuJoCo environment is not entirely new. We extended the modified MuJoCo benchmark introduced in DOMINO (https://arxiv.org/abs/2210.04209) and CaDM (https://arxiv.org/abs/2005.06800). Hence, in our extension, there are four new-added environments (Walker, Crippled Hopper, Crippled Walker, HumanoidStandup) compared with the original benchmark. Additionally, in our experiments, we used a different task set design than in the DOMINO and CaDM papers. Specifically, we evaluated RL algorithms in out-of-distribution tasks which have more extreme environmental features.
>
> For benchmarking, we referred to the DOMINO paper and chose PEARL as a baseline. Our baseline CCM (https://arxiv.org/pdf/2009.13891) is also a zero-shot meta-RL algorithm, and in the CCM paper, CaDM and PEARL were used as baselines for comparison. TESAC, mentioned in the paper (https://arxiv.org/pdf/1910.10897) as a multi-task soft actor-critic with an off-policy version of task embeddings, was used for comparison with the reviewer's recommended $\text{RL}^2$ and MAML. Therefore, we believe our choice of benchmarking aligns with the reviewer's expectations.
>
> Additionally, we directly compared our zero-shot meta-RL algorithms with DOMINO and CaDM using exactly the same task set design. We present the comparison results in Appendix G "A Comparison with DOMINO and CaDM." Since we cannot replicate their pre-trained encoder and prediction networks, we have only compared our results with the experimental results from their original papers. The results show that our model-free methods SaCCM and SaTESAC significantly outperform the model-free DOMINO and are slightly inferior to the model-based CaDM.
>
> **Re Weakness (4):** In the updated manuscript, we have added explanations on how the proposed method works during meta-training and meta-testing times. First, SaNCE is an objective for contrastive learning that can be used to train context encoders. Therefore, we have added details in Section 3, "Preliminaries," under the "Contrastive learning" subsection, on how the context encoder is trained and works during the meta-training and meta-testing phases: "In a Meta-RL setting, the context encoder $\psi(c|\tau_{0:t})$ first takes the trajectory $\tau_{0:t}= \{s_{0}, a_{0}, r_{0},...,s_{t}\}$ from the current episode as input and compresses it into a context embedding $c$. Then, the policy $\pi$, conditioned on context embedding $c$, consumes the current state $s_t$, outputs the action $a_t$. The policy $\pi$ conditioned on a fixed $c$ alters the state of the environment in a consistent way, thereby exhibiting a mode of skill." Furthermore, in Figure 5, we updated the caption and gave a more illustrative description of how the context encoder works during meta-training and meta-testing times. Besides, to help readers understand the learning procedure, we provide pseudo-code for how to use SaNCE during meta-training and meta-testing in Algorithms 1 and 2 in Appendix D.3 (you can find them in the pdf file we submitted in the global rebuttal).
>
> **Re Q(1) how many episodes does the agent interact with the environment:** We have detailed this in Appendix D.3 "Implementation details" under the “Base algorithm” section in the original manuscript. In the Meta-training phase, we trained agents for 1.6 million timesteps in each environment on the Panda-gym and MuJoCo benchmarks. The maximum episode length is listed in Appendix Table 3 (can be found in the pdf file we submitted in the global rebuttal). For meta-testing, we tested 100 episodes in each environment, with tasks randomly sampled from the moderate and extreme task sets. Considering this might be of interest to the readers, we have added this detail in the main paper's Section 5.1 “Experimental Setup” under the “Our Method” paragraph.
>
> **Re Q(2) How does the encoder accumulate across multiple episodes of interaction?:** Our proposed method needs to adapt within an episode. Therefore, at the beginning of each episode, the encoder is reinitialised and does not accumulate across multiple episodes. Specifically, we use an LSTM as the encoder, and we initialise the hidden state and cell state to zero at the start of each episode. We have added the aforementioned content in the “Encoder Architecture” paragraph of Appendix D.3 "Implementation Details" in the updated manuscript.
>
> **Re Q(3) Is there a way to use the proposed method for adaptation within an episode?:** Our proposed method adapts within an episode. We have added descriptions in Section 3, “Preliminaries,” under the “Contrastive learning” subsection and in Figure 5 to make this clearer.
>
> Finally, we have provided a demonstration video, "SaMI.MP4," in our Anonymous GitHub repository (https://anonymous.4open.science/r/SaMI) to better illustrate that our method helps the agent complete the three steps of "explore effectively, infer, adapt" within an episode.

---

> > ### Comment · Reviewer_Qb7x · 2024-08-11
> >
> > Thank you for the response.
> >
> > I don't think it is fair to describe CCM as a zero-shot meta-rl method as it explicitly uses a different policy for first collecting exploration episodes and then runs the exploitation policy given the context encoding.
> >
> > My concerns have been mostly addressed so I'm raising my score.

---

### Official Review · Reviewer_h8bu · 2024-07-10

**Soundness:** 2
**Presentation:** 3
**Contribution:** 2
**Rating:** 6
**Confidence:** 4

**Summary:**

This paper introduces Skill-aware Mutual Information (SaMI) and Skill-aware Noise Contrastive Estimation (SaNCE) to enhance zero-shot generalization in reinforcement learning (RL). The authors address the challenges faced by Meta-Reinforcement Learning (Meta-RL) agents in tasks requiring different optimal skills by using context encoders based on contrastive learning. SaMI distinguishes context embeddings according to skills, while SaNCE, a K-sample estimator, optimizes the SaMI objective, reducing the negative sample space and mitigating the log-K curse. The proposed methods are empirically validated on modified MuJoCo and Panda-gym benchmarks, demonstrating significant improvements in zero-shot generalization and robustness to sample size reductions.

**Strengths:**

### S1. Novel Problem Formulation

The paper introduces a novel and relevant problem formulation by addressing the need for distinguishing context embeddings according to skills in Meta-RL. This approach targets the challenge of generalizing across tasks with varying optimal skills, which is crucial for effective RL in diverse environments.

### S2. Empirical Validation

The proposed methods, SaMI and SaNCE, are thoroughly validated through experiments on modified MuJoCo and Panda-gym benchmarks. The results demonstrate substantial improvements in zero-shot generalization to unseen tasks, showcasing the practical applicability and effectiveness of the methods.

### S3. Clear Presentation

The paper is well-structured, with clear explanations of the proposed methods and their benefits. The use of figures, such as visualizations of context embeddings and success rate comparisons, effectively supports the presentation of results. The theoretical proofs and experimental details provided in the appendices further enhance the clarity and comprehensiveness of the paper.

**Weaknesses:**

### W1. Scalability Concerns

The scalability of the proposed methods to more complex, large-scale environments is not thoroughly discussed. While the methods show promising results in the tested benchmarks, a broader analysis of their scalability and practical utility in more complex scenarios is needed to understand their full potential and limitations.

### W2. Dependency on Skill Definitions

The success of the proposed methods depends on the accurate definition and identification of skills. The paper does not address potential issues related to the variability in skill definitions across different environments and tasks, which could impact the robustness and generalizability of the methods.

**Questions:**

### Q1. Scalability to Complex Environments

How do the proposed methods scale to more complex, large-scale environments? Are there any specific challenges or limitations that need to be addressed for practical deployment in such scenarios?

### Q2. Variability in Skill Definitions

How does the variability in skill definitions across different environments and tasks affect the proposed methods? Have the authors considered any strategies to mitigate the potential impact of inconsistent skill definitions on the robustness and generalizability of the methods?

**Limitations:**

The authors acknowledge the limitations related to the requirement for accurate skill definitions and the focus on specific benchmarks. However, a more detailed discussion on potential negative societal impacts and strategies to mitigate them would be beneficial.

---

> ### Author Rebuttal · Authors · 2024-08-06
>
> Thank you very much for your feedback, especially your comments on the scalability of the method and variability in skill definitions. We are also grateful for the reviewer's recognition of the novelty and contributions of our work, as well as the acknowledgement of our empirical validation and presentation. Please let us know if further clarification is required.
>
> **Re Q1. Scalability to Complex Environments:** To further explore the full potential and limitations of SaMI, we have added 4 new environments in the MuJoCo benchmark (Walker, Crippled Walker, Crippled Hopper, and HumanoidStandup), adopted broader test tasks (with more extreme unseen mass and damping values in testing tasks, shown in Table 3 in the pdf file we submitted in the global rebuttal) and conducted a more comprehensive analysis (i.e., video demos in our Anonymous GitHub repository (https://anonymous.4open.science/r/SaMI) to demonstrate the different skills learned by various algorithms. Through additional experiments, we have gained a clearer understanding of SaMI's scalability to more complex, large-scale environments:
> 1) In more complex MuJoCo environments (Crippled Ant, Crippled Hopper, Crippled Half-Cheetah, SlimHumanoid, HumanoidStandup, and Crippled Walker) as well as the Panda-gym benchmark, SaMI helps the RL agents to be versatile and embody multiple skills, and leads to increased returns/success rates during training and zero-shot generalisation. These environments require the agent to master different skills. When the environmental features (e.g., cube mass, table friction) vary between tasks, the agent can use different skills to work across multiple tasks.
> 2) In the MuJoCo Ant, Half-Cheetah and Hopper environments, we observed that when the environment requires only a single skill, the improvement is not as significant. However, SaMI requires far fewer samples.
> We believe that with the 10 environments in MuJoCo and Panda-gym, we can conclude that SaMI demonstrates significant advantages in complex, large-scale environments. It enables the agent to become more versatile, thereby performing well in these complex settings.
>
> As for the limitations, we discussed that we did not make independence assumptions about environmental features in the Discussion section. Therefore, there is potential to apply our approach to more complex environments where environmental features interact with each other (e.g., the material of the cube affecting both its mass and friction). This will be explored in our future work. We also hope this study inspires others to focus on scalability in complex environments without relying on restrictive assumptions.
>
> We have also added another limitation in Section 5.3 “MuJoCo” under the “Results and Skill Analysis” paragraph, noting that the RL agent tends to learn a single skill to work across multiple tasks in some environments. For example, in the Hopper environment, we found that to adapt to different mass values, the TESAC/CCM/SaTESAC/SaCCM policy tends to learn only one skill, which is the Hopper hopping forward on the floor. As a result, we can see from Table 2 (in the pdf file we submitted in the global rebuttal) that the returns obtained by the four algorithms are very similar across the training tasks, moderate test tasks, and extreme test task settings. In this case, our conclusion is that "even though SaNCE uses fewer samples, it does not degrade RL performance when only one skill is required."  Additionally, further discussion and research are needed to determine when multiple skills are required and what improvements multiple skills can bring beyond generalisation.
>
> **Re Q2. Variability in Skill Definitions:** To more clearly explain the distinctiveness of skills, we have added a description in Section 4.4 "Skill-aware trajectory sampling strategy":
>
> "Methods that focus on skill diversity often rely heavily on accurately defining and identifying skills (https://arxiv.org/abs/1802.06070), with some requiring a prior skill distribution that is often inaccessible (https://arxiv.org/abs/2207.07560). This variability in skill definitions across tasks can affect the robustness and generalisability of these methods. For these reasons, our approach does not use such skill definitions and priors for specific environments or tasks. In this section, we give the distinctiveness of skills and propose a practical trajectory sampling method.
>
> In this study, we believe that distinctiveness of skills is inherently difficult to achieve — a slight difference in states can make two skills distinguishable, and not necessarily in a semantically meaningful way. Instead, we should focus on whether the skills acquired by the agent can complete the task. For example, in high-friction tasks, the agent must acquire the Pick\&Place skill to avoid large frictional forces, while in high-mass tasks, the agent must learn the Push skill since it cannot lift the cube. In that way, without skills definition in a semantically meaningful way, we only need to train an agent on a set of tasks, and it will autonomously discover diverse skills to work across multiple tasks."
>
> Additionally, based on the reviewer's comments, we have added video demos (https://anonymous.4open.science/r/SaMI) in our experiments to help readers better understand how RL agents complete different tasks through different skills. Meanwhile, using the t-SNE and PCA plots provided in the original manuscript, we can understand the distinctiveness of the acquired skills in a semantically meaningful way. For example, in the videos, we can see that SaCCM has learned the Push, Pick\&Place, Drag and Slide skills in Panda-gym.

---

> > ### Comment · Reviewer_h8bu · 2024-08-11
> > **Acknowledgement**
> >
> > I appreciate the authors for the detailed response, especially the video demos, which seem to be very effective.
> >
> > Most of my concerns are addressed, therefore I'm raising my assessment.

---

### Official Review · Reviewer_TjCk · 2024-07-11

**Soundness:** 3
**Presentation:** 3
**Contribution:** 3
**Rating:** 6
**Confidence:** 3

**Summary:**

Learning generalizable skills across different tasks is desirable in Reinforcement Learning. Some methods embed task information into a context latent space which is then used to train policies. This paper proposes a new objective Skill-aware Mutual Information (SaMI) that incentivizes the distinction of context embeddings according to skills. If optimized, the agent can then execute a skill depending on the underlying context. In addition to the objective, the paper proposes a K-sample estimator, Skill-aware Noise Contrastive Estimation (SaNCE), that enables an agent to optimize the SaMI objective in a sample-efficient manner.

**Strengths:**

- relevant topic: Generalising RL agents across tasks is an important topic that will enhance the field of RL
- well-motivated: The example in the introduction in addition to Fig. 1 is helping the reader to visually understand the considered problem
- The paper clearly states the contributions

**Weaknesses:**

only minor things such as an algorithm box would be useful to understand the learning procedure

**Questions:**

- Section 5.3 provides insights into how the positive and negative samples are determined. How prone is this to declare very similar samples, i.e. similar behaviors as positive and negative samples? For example, the return for sample 1 might be high and the return for sample 2 is low, but they are in essence executing the same behavior.

- Could the authors elaborate on the training procedure? How is the sampling of features exactly done and what exactly is the training set as mentioned in Section 6.1 line 246? Does this mean the RL agent is building upon existing offline data?

**Limitations:**

Overall I found this a good paper with minor issues/unclear points (see Weaknesses/Questions).

---

> ### Author Rebuttal · Authors · 2024-08-06
>
> Thank you very much for your recognition of our work. We fully agree with the reviewer that generalising RL agents across tasks is an important topic that will enhance the field of RL, and we are very grateful for your positive feedback on it. We appreciate your acknowledgement of our presentation and our contribution statement. Based on the reviewer's comments, **to help readers understand the learning procedure,** we provide pseudo-code for how to use SaNCE during meta-training and meta-testing in Algorithms 1 and 2 in Appendix D.3 "Implementation Details" (you can find them in the pdf file we submitted in the global rebuttal). Please let us know if further clarification is required.
>
> **Re Q(1) “How prone is this to declare very similar samples, i.e. similar behaviours, as positive and negative samples? For example, the return for sample 1 might be high and the return for sample 2 is low, but they are in essence executing the same behaviour”:** In Section 5.3, we define samples (i.e., trajectories) with high returns as positive samples, and the skill that generates this positive sample is defined as a positive skill. In this way, if sample 1 and sample 2 do not have similar returns (i.e., sample 1 has a high return, and sample 2 has a low return), then their corresponding skills are also not similar (i.e., skill 1 is a positive skill, and skill 2 is a negative skill). Note that the similarity of skills mentioned here is not semantic similarity; it does not mean that all skills that exhibit "pick the cube off the table and place it to the goal position" are considered similar skills. Furthermore, the reviewer's concern involves the issue of the distinctiveness of skills. To more clearly explain the distinctiveness of skills, we have added a description in Section 4.4 "Skill-aware trajectory sampling strategy":
>
> "Methods that focus on skill diversity often rely heavily on accurately defining and identifying skills (https://arxiv.org/abs/1802.06070), with some requiring a prior skill distribution that is often inaccessible (https://arxiv.org/abs/2207.07560). This variability in skill definitions across tasks can affect the robustness and generalisability of these methods. For these reasons, our approach does not use such skill definitions and priors for specific environments or tasks. In this section, we give the distinctiveness of skills and propose a practical trajectory sampling method.
>
> In this study, we believe that distinctiveness of skills is inherently difficult to achieve — a slight difference in states can make two skills distinguishable, and not necessarily in a semantically meaningful way. Instead, we should focus on whether the skills acquired by the agent can complete the task. For example, in high-friction tasks, the agent must acquire the Pick\&Place skill to avoid large frictional forces, while in high-mass tasks, the agent must learn the Push skill since it cannot lift the cube. In that way, without skills definition in a semantically meaningful way, we only need to train an agent on a set of tasks, and it will autonomously discover diverse skills to work across multiple tasks."
>
> **Re Q(2): “How is the sampling of features exactly done and what exactly is the training set as mentioned in Section 6.1, line 246? Does this mean the RL agent is building upon existing offline data?”:** The original text in line 246 is “During training, we uniform-randomly select a combination of environmental features from a training set.”
> This sentence means that during meta-training, at the beginning of each episode, we sample a task from the training task set to interact with for one episode. Since tasks are defined by environmental features, this means selecting a combination of environmental features from a training task set at the start of each episode. To ensure it is clear to readers that "training set" refers to data collected during training rather than offline data, we have revised the original text to: “During meta-training, we uniform-randomly select a combination of environmental features from a training task set.” Similarly, during meta-testing, at the beginning of each episode, we select a combination of environmental features from the moderate/extreme task set.
>
> Additionally, in response to the reviewer's comment, “Could the authors elaborate on the training procedure?”, we have added the description of the meta-training and meta-testing processes in the new manuscript. We have added descriptions in Section 3, “Preliminaries,” under the “Contrastive learning” subsection and in Figure 5. We also provided an example in the first paragraph of the introduction section illustrating the need for the agent to adapt within an episode: “When facing an unknown environment, the agent needs to explore effectively, understand the environment, and adjust its behaviour accordingly within an episode. For instance, if the agent tries to push a cube across a table covered by a tablecloth and finds it ‘unpushable,’ it should infer that the table friction is relatively high and adapt by lifting the cube to avoid friction, rather than continuing to push.”

---

> > ### Comment · Reviewer_TjCk · 2024-08-12
> >
> > I highly appreciate the authors' responses that clarify my questions. The responses confirm my initial assessment in recommending accepting this paper.

---

### Official Review · Reviewer_hnsz · 2024-07-19

**Soundness:** 1
**Presentation:** 1
**Contribution:** 2
**Rating:** 3
**Confidence:** 4

**Summary:**

The paper proposes an alternative approach to infoNCE for learning contrastive task representations by using the structure of skills in a meta learning setting. In doing so, the algorithm samples more negatives within the same task as the positive (but with lower reward), in a procedure that somewhat resembles hard negative mining. A benefit to this procedure is it is less sensitive to the choice of infoNCE batch size K, which in large multi-task settings is often only able to approximate mutual information for infeasibly large values. Empirical results show this alternative contrastive loss improves performance on RL benchmarks.

**Strengths:**

- The log K curse is an important practical issue when using contrastive infoNCE losses in RL settings with many tasks—often the classification becomes trivial without hard negatives. Approaches like those proposed in this paper will be important to scaling contrastive learning methods for control.
- Empirical results show strong benefits against the baselines tested in many environments.

**Weaknesses:**

- There seem to be issues with the mathematical formulation (see questions)
- I found the presentation of the method to be confusing, in particular regarding how the learned representations were actually used for meta learning

**Questions:**

- I'm confused by the statement in Lemma 1. The RHS, $\log K  \le I(x;y)$ only depends on the data distribution. In some cases, it seems the mutual information between $x$ and $y$ can be less than $\log K$, regardless of the parameterization used. Is this a contradiction?
  - Similarly, I don't understand Figure 6—K is on the x-axis, but $\log K$ is drawn as a horizontal line.

- From eq. 2, it seems the definition of the SaMI $I_{\mathrm{SaMI}}(c ; \pi_c ; \tau_c)=\mathbb{E}_{p(c, \pi_c, \tau_c)} \{\log \frac{p(\tau_c \mid c, \pi_c)}{p(\tau_c)}\}$ is equivalent to the mutual information $I(\tau_c;(c,\pi_c))$.

  But by the chain rule we know $I(\tau_c;(c,\pi_c))=I(\tau_c;c)+I(\tau_c;\pi_c\mid{}c)$ which contradicts the next statement $I_{\mathrm{SaMI}}(c ; \pi_c ; \tau_c) \leq I(c ; \tau_c)$

- How is the "momentum encoder" parameterized in this setting?

- Is there significance to the bars not lining up in Figure 2 (c)?

- The authors may wish to discuss related work on hard negative mining

**Limitations:**

Addressed

---

> ### Author Rebuttal · Authors · 2024-08-06
>
> Thank you for the reviewer's feedback and for acknowledging our empirical results and recognising the importance of our research contributions to scaling contrastive learning methods for control. We hope our clarifications, corrections and analysis can convince the reviewer to reconsider their score. Please let us know if further clarification is required.
>
> **Re Q(1) the statement in Lemma 1 and typo in Figure 6:** Thank you for pointing out the typo in Figure 6 in the original paper, we’ve fixed it in the updated paper, including:
> 1) The x-axis should be "training epochs" instead of $K$. The main idea that Figure 6 aims to present is that as the training progresses, the lower bound of $I(x;y)$, $I_\text{SaNCE}$ (yellow line), gradually approaches $I(x;y)$ (dark blue line).
> 2) Considering the importance of Figure 6 to Lemma 1, we have moved it to the main text (so it's Figure 3 now, you can find it in the one-page pdf).
>
> Regarding the reviewer's question, "In some cases, it seems the mutual information between $x$ and $y$ can be less than $\log K$, regardless of the parameterization used." We have revised the statement of Lemma 1 in Section 4.1 to ensure we have considered all cases. We believe the following two changes will address the reviewer's concern:
>
> 1) We do not consider the case when $x \perp y$, i.e. $\log K > I(x;y) = 0$ ($\forall K \geq 1$), because a meta-RL agent learns a context encoder by maximising MI between trajectories $\tau_c$ and context embeddings $c$, which are not independent according to the MDP graph in Figure 2. We also added the aforementioned content in Section 4.1. More specifically, in Section 4.2, we define skills as follows: “A policy $\pi$ conditioned on a fixed context embedding $c$ is defined as a skill $\pi(\cdot|c)$, abbreviated as $\pi_c$. If a skill $\pi_c$ is conditioned on a state $s_t$, we can sample actions $a_t \sim \pi(\cdot|c,s_t)$. After sampling actions from $\pi_c$ at consecutive timesteps, we obtain a trajectory $\tau_{c}=\{s_t, a_t, r_t, s_{t+1}, \ldots, s_{t+T}, a_{t+T}, r_{t+T}\}$ which demonstrates a consistent mode of behaviour.” Therefore, trajectories $\tau_c$ and context embeddings $c$ are not independent.
>
> 2) Given that we focus on generalisation, we consider cases with a finite sample size of $K$, hence the true $I(x;y) \geq \log K$. This is because we want to learn a context encoder to compress helpful information even with finite samples, which is crucial for generalisation. The key to achieving good compression is having a reliable measure of compression, and MI is a good measure of compression  (https://arxiv.org/pdf/1810.05728). $K$-sample estimators like NCE or InfoNCE were originally proposed to estimate MI from finite samples. Therefore, we believe it is important to consider whether the $K$-sample estimator can approximate MI with finite samples, i.e., we should consider cases where the true $I(x;y) \geq \log K$. We also added the above statement in Section 4.1 on page 4 of the updated manuscript.
>
> We have included the restrictions of "finite sample size $K$" and the independence of $x$ and $y$ in Lemma 1 to ensure that the Lemma covers all cases (as shown on page 4 of the updated manuscript):
> Lemma 1: Learning a context encoder $\psi$ with a $K$-sample estimator and finite sample size $K$, we always have ${I}_{\text{InfoNCE}}(x;y|\psi,K)$ $\leq$ $\log K$ $\leq$ $I(x;y)$, when $x \not \perp y$. (see proof in Appendix A)
>
> **Re Q(2) definition of SaMI in eq.2:** Thank you for pointing out our typo "$p(\tau_c|c,\pi_c)/p(\tau_c)$" in eq.2. We have corrected it, and it now aligns with the formula in Appendix B "Proof for Lemma 2". We have corrected eq. 2 to:
> $$I_{\text{SaMI}}(c;\pi_c;\tau_c) = E_{p(c,\pi_c,\tau_c)}  \log \ \frac{p(c,\pi_c,\tau_c)}{p(c)p(\pi_c)p(\tau_c)} $$
> Besides, we gave proof in Appendix B of the original manuscript: $I_{\text{SaMI}}(c;\pi_c;\tau_c) = I(c;\tau_c) - I(c;\tau_c|\pi_c)$, hence $I_{\text{SaMI}}(c;\pi_c;\tau_c) \leq I(c;\tau_c)$.
>
> **Re Q(3) How is the "momentum encoder" parameterized?:** In Appendix D.3 "Implementation Details" of the original manuscript, the "momentum encoder" is updated $\theta_{\psi^*}$ by $\theta_{\psi^*} \leftarrow m \cdot \theta_{\psi} + (1-m) \cdot \theta_{\psi^*}$. This is a similar setup to existing research (e.g. https://arxiv.org/abs/1911.05722). The momentum update makes $\theta_{\psi^*}$ evolve more smoothly by having it slowly track $\theta_{\psi}$ with $m \ll 1$ (e.g., $m=0.05$ in this research).
>
> **Re Q(4) Is there significance to the bars not lining up in Figure 2(c)?:** Yes, the bars not lining up in Figure 2 (c) (Figure 4 in the updated manuscript) illustrate that the negative sample space may differ across tasks. To address the reviewer's concern, we have added an explanation in Appendix C, "Sample Size of $I_{\text{Sa+InfoNCE}}$,": "Figure 4 illustrates the differences in the size of the negative sample space, in which the negative sample space may vary across tasks (as shown by the non-lining-up bars in Figure 4(c)). This is because we define negative samples as trajectories with low returns, so the size of the negative sample space is influenced by sampling randomness."
>
> **Re Q(5) related work on hard negative mining:** We thank the reviewer for pointing out this related work, such methods provide insight into defining positive/negative samples. We have carefully considered this suggestion, but our method does not make significant contributions in terms of sample hardness. In the third paragraph of Section 4.4 “Skill-aware trajectory sampling strategy,” we briefly added insight we gain from hard negative samples: “Positive samples are generated by the optimal skill for the current task, while lower return samples are classified as negative. This polarised definition helps the model select the optimal skill from among many skills with varying returns and avoids the issue of hard negative examples during training (https://arxiv.org/abs/2010.04592).”

---

> > ### Comment · Reviewer_hnsz · 2024-08-12
> > **Response**
> >
> > Thank you for your response. The new statement of eq. (2) seems to be equivalent to the definition of the *total correlation* (Watanabe, 1960). Could you comment on the relationship between the updated eq. (2) definition of $I_{\\mathrm{SaMI}}$ and existing information-theoretic quantities like total correlation (and estimators such as Bai et al. (2023))?
> >
> >
> > ### References
> > Bai, K., Cheng, P., Hao, W., Henao, R., & Carin, L. (2023). Estimating total correlation with mutual information estimators. _Proceedings of The 26th International Conference on Artificial Intelligence and Statistics_, 2147–2164. [https://proceedings.mlr.press/v206/bai23a.html](https://proceedings.mlr.press/v206/bai23a.html)
> >
> > Watanabe, S. (1960). Information theoretical analysis of multivariate correlation. _IBM Journal of Research and Development_, _4_, 66–82. IBM Journal of Research and Development. [https://doi.org/10.1147/rd.41.0066](https://doi.org/10.1147/rd.41.0066)

---

> > > ### Author Response · Authors · 2024-08-13
> > > **Response to the official comment from Reviewer hnsz**
> > >
> > > **Re “The new statement of eq. (2) seems to be equivalent to the definition of the total correlation”:** We thank the reviewer for pointing out this related work. The eq. (2) $I_{\text{SaMI}}(c;\pi_c;\tau_c)$ is the **interaction information**, and it differs from the **total correlation** in definition. Both interaction information and total correlation are generalisations of mutual information. Specifically, let's first consider the **total correlation** for three variables. For a given set of 3 random variables $\{x,y,z\}$, the total correlation $TC(x,y,z)$ is defined as the Kullback–Leibler divergence from the joint distribution $p(x,y,z)$ to the independent distribution $p(x)p(y)p(z)$. This divergence simplifies to the difference of entropies:
> > > $$TC(x,y,z)=H(x)+H(y)+H(z)-H(x,y,z)$$
> > > where $H(x)$ is the information entropy of variable $x$, and $H(x,y,z)$ is the joint entropy of the variable set $\{x,y,z\}$.
> > >
> > > In contrast, the **interaction information** $I(x;y;z)$ for three variables $\{x,y,z\}$ is given by:
> > > $$I(x;y;z)=(H(x)+H(y)+H(z)) - (H(x,y)+H(x,z)+H(y,z)) + H(x,y,z)$$
> > > In Appendix B, “Proof for Lemma,” we refer to it as “interaction information,” drawing from references "Interaction Information for Causal Inference: The Case of Directed Triangle" (https://arxiv.org/pdf/1701.08868) and "Multivariate Information Transmission" (https://ieeexplore.ieee.org/document/1057469). There are several other names for Interaction Information, including the amount of information [1], information correlation [2], co-information [3], and simply mutual information [4].
> > > To make eq. (2) clearer, we have moved the definition of Interaction Information into the main text. Therefore, we directly defined $I_{SaMI}$ using the Interaction Information definition:
> > > $$I_{SaMI}(c;\pi_c;\tau_c)= I(c;\tau_c)-I(c;\tau_c|\pi_c)$$
> > > Therefore, we proposed a lower bound $I_{\text{SaMI}}(c;\pi_c;\tau_c)$ to approximate $I(c;\tau_c)$. We believe that other generalisations of mutual information (i.e., MI estimators) could also be helpful in approximating $I(c;\tau_c)$.
> > >
> > > **Re "the relationship between the updated eq. (2) definition of $I_{SaMI}$ and estimators such as Bai et al. (2023)":** TC estimators [5] is an interesting study, we fully agree with Bai et al.'s perspective [5] that when calculating mutual information among multiple variables, existing work often seeks ways to decompose interaction information, which leads to too much strong independence assumptions about the data distribution. For instance, as mentioned in [5], some studies make a strong assumption that all variables are conditionally independent [6].  Without such strong independence assumptions, TC estimator [5] is a method of decomposing interaction information and assumes that the relationships between variables follow a tree-like or line-like structure, thus applying tree-like or line-like decomposition to the interaction information, resulting in tree-like or line-like MI estimators (i.e., TC estimators).
> > >
> > > Our proposed SaMI approach does not make any assumptions about the data, and therefore does not decompose the interaction information. This is primarily because we focus on mutual information among just three variables in the context of the RL problem, which is significantly simpler than the scenarios addressed in [5]. Additionally, through extensive experiments, we have verified that maximising the mutual information $I_{SaMI}$ significantly enhances the zero-shot generalisation ability of reinforcement learning.
> > >
> > > We thank the reviewer again for noting the typo in our original eq(2) and helping us to refine Lemma 1. We would be grateful if the reviewer could reevaluate their score in light of our given clarifications.
> > >
> > >
> > > **References**
> > >
> > > [1] Ting, Hu Kuo. "On the amount of information." Theory of Probability \& Its Applications 7.4 (1962): 439-447.
> > >
> > > [2] Wolf, David R. "The Generalization of Mutual Information as the Information between a Set of Variables: The Information Correlation Function Hierarchy and the Information Structure of Multi-Agent Systems." (2004).
> > >
> > > [3] Bell, Anthony J. "The co-information lattice." Proceedings of the fifth international workshop on independent component analysis and blind signal separation: ICA. Vol. 2003. 2003.
> > >
> > > [4] Yeung, Raymond W. "A new outlook on Shannon's information measures." IEEE transactions on information theory 37.3 (1991): 466-474.
> > >
> > > [5] Bai, Ke, et al. "Estimating total correlation with mutual information estimators." International Conference on Artificial Intelligence and Statistics. PMLR, 2023.
> > >
> > > [6] Poole, Ben, et al. "On variational bounds of mutual information." International Conference on Machine Learning. PMLR, 2019.

---

> > > > ### Comment · Reviewer_hnsz · 2024-08-13
> > > >
> > > > Thank you for your response. In your first comment, you stated $I_{\mathrm{SaMI}}\left(c ; \pi_c ; \tau_c\right)=E_{p\left(c, \pi_c, \tau_c\right)} \log \frac{p\left(c, \pi_c, \tau_c\right)}{p(c) p\left(\pi_c\right) p\left(\tau_c\right)}$, while in the second you state $I_{\text {SaMI }}\left(c ; \pi_c ; \tau_c\right)=I\left(c ; \tau_c\right)-I\left(c ; \tau_c \mid \pi_c\right)$. The first definition is total correlation while the second is 3-variable interaction information. Is the second statement the correct definition then? I am still confused as to which form is optimized by your objective and the motivation for maximizing interaction information.

---

> ### Author Response · Authors · 2024-08-13
> **Response to the official comment from Reviewer hnsz**
>
> Thank you for your response. **The second statement is the correct definition $I_{SaMI}(c;\pi_c;\tau_c)= I(c;\tau_c)-I(c;\tau_c|\pi_c)$, i.e., the 3-variable interaction information.** So we are optimising the interaction information $I_{SaMI}(c;\pi_c;\tau_c)$, defined as $I_{SaMI}(c;\pi_c;\tau_c) = I(c;\tau_c)-I(c;\tau_c|\pi_c)$. Please kindly check the updated manuscript for the updated eq. (2) at https://anonymous.4open.science/r/SaMI/SaMI.pdf.
>
> So the entire story of the version changes is as follows:
>
> 1) **The SaMI formula $I_{SaMI}(c;\pi_c;\tau_c)$, defined based on interaction information, has not changed,** and the proof (in Appendix A and B of our original manuscript) based on this formula still holds.
>
> 2) **The only change made was to the detailed version of the formula with the log probabilities, i.e., $E_{p(c,\pi_c,\tau_c)}  \log \ \frac{p(\tau_c|c,\pi_c)}{p(\tau_c)}$.** This was corrected due to a typo pointed out by the reviewer, but it was not essential for the theory and has been removed from the paper. Since we cannot evaluate $p(c,\pi_c,\tau_c)$ directly, we approximate it by Monte Carlo sampling using $K$ samples from $p(c,\pi_c,\tau_c)$ (i.e., trajectories from environments). Therefore, extensive discussion about the details of the distribution $p(c,\pi_c,\tau_c)$ is not required.
>
> 3) We ensure that our framework effectively maximises $I_{SaMI}$ through the $K$-sample estimator $I_{SaNCE}$ proposed in Section 4.3 of the original manuscript, and the skill-aware trajectory sampling strategy proposed in Section 4.4  of the original manuscript.
>
>
> **Please refer to Section 4.1 of our original manuscript for our motivation.** Since maximising MI $I(c;\tau_c)$ faces the $\log K$ curse, we propose $I_{\text{SaMI}}(c;\pi_c;\tau_c)$, which is a ground-truth MI that is smaller than $I(c;\tau_c)$. We state in our Lemma 1 that $I_{\text{InfoNCE}}(x;y|\psi, K) \leq \log K \leq I(x;y)$; please refer to Appendix A of our original manuscript for the proof of Lemma 1.
>
> We have presented **the motivation for this study in Section 4.1 of our original manuscript:** “Good compression is crucial for generalisation, and compressing valuable information from a limited number of $K$ samples requires MI as an effective measure of compression (https://arxiv.org/abs/1810.05728). We derive three key insights when learning a context encoder with finite sample size: (1) focus on a ground-truth MI that is smaller than $I(x;y)$; (2) develop a $K$-sample estimator tighter than the $I_{\text{InfoNCE}}$; (3) increase sample quantity $K$, however, this is usually impractical. A meta-RL agent learns a context encoder by maximizing MI between trajectories $\tau_c$ and context embeddings $c$. Driven by insight (1), we introduce Skill-aware Mutual Information (SaMI) in Section 4.2, designed to enhance the zero-shot generalization of downstream RL tasks. Corresponding to insight (2), we propose Skill-aware Noise Contrastive Estimation (SaNCE) to maximize SaMI with finite samples in Section 4.3. Finally, Section 4.4 demonstrates how to equip a Meta-RL agent with SaNCE in practice.”
>
> Besides, there are many formulas (or generalised versions of these formulas) for calculating the mutual information between variables $c,\pi_c,\tau_c$. **We aim to address the $\log K$ curse. Therefore, we proposed $I_{SaMI}(c;\pi_c;\tau_c)$, which aligns with our motivation: $I_{SaMI}$ serves as a ground-truth MI that is smaller than $I(c;\tau_c)$. Hence, we maximise the interaction information $I_{SaMI}$.** This allows for faster convergence in sample-limited tasks, thereby facilitating zero-shot generalisation. Achieving this is crucial for applying RL-based robots to sample-limited real-world scenarios and enabling rapid adaptation, which is one of the significant bottlenecks.
>
> We believe there are various bounds of mutual information, lower bounds and upper bounds (https://arxiv.org/pdf/1905.06922). For example, **the total correlation pointed out by the reviewer has great potential for calculating mutual information among multiple variables. However, it does not align with our motivation.** Since $TC \geq I_{SaMI}$, it is not suitable for zero-shot generalisation scenarios.

---

### Official Review · Reviewer_cm3M · 2024-07-25

**Soundness:** 3
**Presentation:** 3
**Contribution:** 3
**Rating:** 6
**Confidence:** 3

**Summary:**

The paper presents an interesting way to deal with the "log k curse", that is novel while being sensible. The results seem promising, even though their presentation needs improving, and I believe it can serve as foundation for many future works to build upon. Overall I think the idea and execution are solid, but the paper as an artifact needs work, as I'll point out in detail in the Weaknesses section.

**Strengths:**

The paper presents an interesting novel technique that is intuitive without being trivial. There are experiments across 2 suites of tasks and solid mathematical work, and the overall idea is flexbible enough for future work to build upon.

**Weaknesses:**

I believe the results, and more specifically their presentation is a serious weakness of the paper. In Panda-Gym the authors claim their method "boosts success rates by an average of 20.23% in moderate tests and18.36% in extreme tests", but in fact when taking into account the standard deviation of theirs and previous method performances we *cannot* ascertain whether there was any improvement whatsoever on the moderate and extreme tests. This also points to an abuse of the use of the blue background to "indicates that our method surpasses PEARL, TESAC and CCM".

Moreover in the MuJoCo results while we in general do see more statistically significant results for Test (Moderate) and Test (extreme) there are still a few misuses of the blue background, and also we in general don't see statistically significant improvement in the training setting, which leads to the question of why do we see such different behaviours in Panda-Gym (non-statistically significant improvement in Moderate and Extreme, but improvement in Training) and MuJoCo (the opposite), a question that would have been good if the authors had explored a little more.

Finally the interpretation of the t-SNE results seem quite problematic, where 2 clear clusters, one on the lower left and another on the lower right, containing context embeddings of the 4 kinds of settings for Panda-Gym seem to be ignored, but an artificial cluster is suggested by using a yellow bounding box. I consider such cluster artificial as  many of the points on the upper part of the bounding seem closer to points outside it then to points inside it, and it's always good to remember that for t-SNE local distances are meaningful, but global ones are not.

**Questions:**

+ Do the authors have any hypothesis on why the generalization behaviour is different when using SaNCE in Panda-Gym vs MuJoCo?
+ I believe fixing the plots and claims on improvements is fundamental for the paper to be a good research artifact and recommend the authors do so
+ Given what was said about the t-SNE analysis I believe it either should simply be removed, or a different analysis using clustering methods in higher dimensions should be used instead.

**Limitations:**

The paper is mostly about addressing a limitation of previous methods, and I do not believe it has any potential negative societal impact.

---

> ### Author Rebuttal · Authors · 2024-08-06
>
> Thank you for the reviewer’s comments, especially the helpful comments on the presentation of experimental results. We appreciate your recognition of the novelty of our work and the potential of our method. We have made revisions to enhance the clarity and presentation of our experimental results based on your suggestions. Please let us know if further clarification is required.
>
> **Re statistical significance tests:** We have added a t-test to conduct a statistical hypothesis test to determine whether SaMI brought a statistically significant improvement over the respective base RL algorithm without SaMI. i) In the updated manuscript, we marked with an asterisk (*) in Table 1 and Table 2 where the algorithm with SaMI shows statistically significant improvement over the same algorithm without SaMI (you can also find them in the pdf file we submitted in the global rebuttal). The t-test results indicate that SaMI brings significant improvement to the extreme test set. This aligns with our conclusions, that SaMI leads to increased returns during training and zero-shot generalisation, especially in environments that require different skills. We have added this information in Section 5.3. ii) We also report the p-values of the t-tests in Appendix H in the updated manuscript.
>
> **Re Q(1) whether the generalisation behaviour is different for SaNCE in Panda-Gym vs MuJoCo:** The results from Panda-Gym and MuJoCo are consistent as they both show: i) SaMI helps the RL agents to be versatile and embody multiple skills; ii) SaMI leads to increased performance during training and zero-shot generalisation, especially in environments that require different skills. Specifically,
>
> *1) Panda-Gym:* The RL agent must equip itself with different skills to achieve a high success rate across multiple tasks because different environmental features require different skills. For example, tasks with high friction require the Pick\&Place skill (picking the cube off the table and placing it in the goal position), while tasks with high mass require the Push skill (pushing the cube to move it across the table). Therefore, in the Panda-Gym benchmark, we see that multiple skills are needed in training tasks, moderate test tasks, and extreme test tasks, so SaNCE brings improvement across all three.
>
> *2) MuJoCo:* In the MuJoCo environment, we found:
> i) In some environments (Ant, Hopper, and Half-Cheetah), the RL agent only needs to learn a single skill to generalise across different tasks. For example, in the Hopper environment, we found that to adapt to different mass values, the TESAC/CCM/SaTESAC/SaCCM policy tends to learn only one skill, which is the Hopper hopping forward on the floor. As a result, we can see from Table 2 that the returns obtained by the four algorithms are very similar across the training tasks, moderate test tasks, and extreme test task settings. In this case, our conclusion is that "even though SaNCE uses fewer samples, it does not degrade RL performance";
> ii) When the environment (Crippled Ant, Crippled Hopper, Crippled Half-Cheetah, SlimHumanoid, HumanoidStandup, and Crippled Walker) requires different skills to complete tasks, SaNCE brings significant improvements. For example, in the Crippled Ant environment, when it has 3 or 4 legs available, the Crippled Ant Robot learns to roll to adapt to varying mass and damping in the training tasks and moderate tasks. However, during zero-shot generalisation in extreme tasks, when only 2 legs are available, the Ant Robot can no longer roll. Instead, it adapts by walking using its two legs. Therefore, we can see that SaNCE brings significant improvement, especially during zero-shot generalisation.
>
> To better showcase generalisation behaviour, we have added 4 new environments in the MuJoCo benchmark (Walker, Crippled Walker, Crippled Hopper, and HumanoidStandup), and adopted broader test tasks (with more extreme unseen mass and damping values in testing tasks, shown in Table 3 in the pdf file we submitted in the global rebuttal). This allows us to better demonstrate SaNCE's zero-shot generalisation ability across more environments and less familiar testing tasks. Additionally, to better present our results, we have utilised videos on GitHub to clearly demonstrate how SaNCE helps the RL algorithm learn different skills and perform zero-shot generalisation (https://anonymous.4open.science/r/SaMI).
>
> To help readers better understand the generalisation behaviour, we have added the aforementioned descriptions in Section 5.2 "Panda-gym" and Section 5.3 "MuJoCo" under the "Result and Skill Analysis" paragraph.
>
> **Re Q(2) and (3) t-SNE plots and claims on improvements:** We have updated the t-SNE plots in Figure 6 in the one-page pdf. The original plot represented 100 trajectories for each task, where each point in the t-SNE plot represents the context embedding of the final time step of each trajectory, encapsulating the skill information of the entire trajectory. However, when randomly initialising the cube's starting position, if the cube is very close to the target position, the agent only needs to slightly move the cube to complete the task. In this case, the embeddings for all 4 tasks cluster together. Additionally, there are failure cases across all 4 tasks where the agent attempts to move the cube, but the cube remains still, leading to clustering of the embeddings. Therefore, the 2 clear clusters containing all tasks represent these two scenarios, with one cluster on the lower left and another on the lower right. To better illustrate the clustering of embeddings based on different skills, we have removed the aforementioned two scenarios from the 100 tests for each task. In the revised t-SNE plot, we can clearly see two distinct clusters.
>
> **Regarding the presentation of results:** To better present our experimental results, we have made the following improvements: 1) We’ve deleted the blue background. 2) We have removed the average improvement percentage.

---

> > ### Comment · Reviewer_cm3M · 2024-08-10
> >
> > Thank you for your response and the changes made to the paper.
> > Could you tell me the 5 values for the performances of SaTESAC and TESAC on the Panda-Gym extreme test setting? I'm somewhat confused by the t-test results.
> > I still do not understand how you can call the region inside the yellow box a cluster in T-SNE, as there are multiple points with x < -10 who look like they should be part of the so-called cluster, but which would be pretty bad for the push skill, which end up just making me doubt a little whether the method is truly doing what the authors claim it is doing.

---

> > > ### Author Response · Authors · 2024-08-11
> > > **Response to the official comment from Reviewer cm3M**
> > >
> > > Thank you for your response. The table below provides the requested 5 values for the performances of SaTESAC and TESAC showing the average success rate ± standard deviation (on 8 extreme test tasks):
> > >
> > > |             | seed 1 | seed 2  | seed 3 | seed 4  | seed 5 |
> > > | -------- | ------- | ------- | ------- | ------- | ------- |
> > > | TESAC  | 0.19±0.13 | 0.20±0.26 | 0.25±0.19 | 0.18±0.14 | 0.28±0.31 |
> > > | SaTESAC | 0.37±0.37 | 0.37±0.35 | 0.36±0.36 | 0.36±0.34 | 0.38±0.35 |
> > >
> > > These 5 values in the table demonstrate that *the improvements made by SaTESAC over TESAC in extreme test tasks are significant*, which aligns with our paired t-test results. The paired t-test indicates that *the improvements made by SaTESAC over TESAC in all three settings (Training, Moderate, and Extreme) are statistically significant at a significance level of 0.05*. The paired t-test results are shown in the table below (you can also find these results in Appendix H of the updated paper at: https://anonymous.4open.science/r/SaMI/SaMI.pdf):
> > >
> > > |                   | Training  | Moderate  | Extreme   |
> > > |-------------------|-----------|-----------|-----------|
> > > | t-statistic       | 12.180000 | 13.800000 | 8.020000  |
> > > | p-value    | 0.000260 | 0.000160 | 0.001310 |
> > >
> > > Please find all the results for TESAC/SaTESAC/CCM/SaCCM across multiple seeds used to generate Table 1 and Table 2 in our anonymous GitHub repository (https://anonymous.4open.science/r/SaMI/data/MuJoCo.xlsx and https://anonymous.4open.science/r/SaMI/data/Panda-gym.xls). Please refer to the "success rate \& t-test" sheet in the file “data/Panda-gym.xls” to find all experimental results related to Panda-gym. The content of the table above can also be found in the "Mean Success Rate-Multiple Envs" sheet.
> > >
> > > Thank you for your comment on the projection method. We have added **UMAP** (https://arxiv.org/pdf/1802.03426) as an alternative projection method, which is similar to t-SNE but more efficient and tends to better preserve the global structure of the data than t-SNE (i.e., more clearly separates groups of similar categories from each other). We have replaced the t-SNE visualisation in Figure 6 with UMAP in the updated paper. In the new context embedding, two clear clusters are visible, corresponding to Push and Pick\&Place, respectively. Please find the updated Figure 6 at https://anonymous.4open.science/r/SaMI/data/figure_6.png
> > >
> > > Finally, to better understand what the method is doing, combining the newly added UMAP visualisations (in Appendix F of the updated paper at: https://anonymous.4open.science/r/SaMI/SaMI.pdf) with Heatmap visualisations (provided in the original manuscript in Appendix F), and video demos (in our anonymous GitHub repository https://anonymous.4open.science/r/SaMI) would be helpful. We have provided skill analysis in the original manuscript in sections 5.2 and 5.3 under the “Results and skill analysis” paragraph, which you can also find on page 9 (Panda-gym environment) and page 10 (MuJoCo environment) of the updated manuscript (https://anonymous.4open.science/r/SaMI/SaMI.pdf).
> > >
> > > Please let us know if further clarification is required.

---

> > > > ### Comment · Reviewer_cm3M · 2024-08-12
> > > >
> > > > The new UMAP plot indeed shows the clustering you allude to. Given that you addressed my comments I'll increase my rating

---

### Author Rebuttal · Authors · 2024-08-06

Thank you very much for the reviewer's feedback. We appreciate the time and effort that the reviewers dedicated to providing feedback on our manuscript, and are grateful for the insightful comments and valuable improvements to our paper. We have addressed each of the reviewers' comments individually. Please find video demos in our Anonymous GitHub repository (https://anonymous.4open.science/r/SaMI). We also attach a one-page PDF file which contains all of the Tables and Figures mentioned in our responses to the reviewers. You can also find our updated anonymous manuscript in our Anonymous GitHub repository (named "SaMI.pdf").

---

### Decision · Program_Chairs · 2024-09-25

**Decision:**

Accept (poster)

**Comment:**

The submission clearly presents a novel method with sufficient empirical evidence for its efficacy. There were some concerns raised in the reviews, but most (if not all) of them were adequately addressed by the authors during the rebuttal.

As such, I am recommending an acceptance.